## PROCEEDINGS A

statistics, mathematical modelling, computational mathematics

partially observed diffusions, randomization methods, Hessian estimation, coupled conditional particle filter

**Author for correspondence:**
Neil K. Chada
e-mail: neilchada123@gmail.com

# Unbiased estimation of the Hessian for partially observed diffusions

Neil K. Chada, Ajay Jasra and Fangyuan Yu

Computer, Electrical and Mathematical Sciences and Engineering Division, King Abdullah University of Science and Technology, Thuwal, 23955-6900, Saudi Arabia

NKC, 0000-0002-2180-0985

In this article, we consider the development of unbiased estimators of the Hessian, of the log-likelihood function with respect to parameters, for partially observed diffusion processes. These processes arise in numerous applications, where such diffusions require derivative information, either through the Jacobian or Hessian matrix. As time-discretizations of diffusions induce a bias, we provide an unbiased estimator of the Hessian. This is based on using Girsanov's Theorem and randomization schemes developed through Mcleish (2011 *Monte Carlo Methods Appl.* **17**, 301–315 (doi:10.1515/mcma.2011.013)) and Rhee & Glynn (2016 *Op. Res.* **63**, 1026–1043). We demonstrate our developed estimator of the Hessian is unbiased, and one of finite variance. We numerically test and verify this by comparing the methodology here to that of a newly proposed particle filtering methodology. We test this on a range of diffusion models, which include different Ornstein–Uhlenbeck processes and the Fitzhugh–Nagumo model, arising in neuroscience.

## 1. Introduction

In many scientific disciplines, diffusion processes [1] are used to model and describe important phenomena. Particular applications where such processes arise include biological sciences, finance, signal processing and atmospheric sciences [2–5]. Mathematically, diffusion processes take the general form

$$\mathrm{d}X_t = a_\theta(X_t)dt + \sigma(X_t)\,\mathrm{d}W_t, \quad X_0 = x_\star \in \mathbb{R}^d, \qquad (1.1)$$

**Figure 1.** Simulation of FHN model for $t = 50$. Purple crosses in the top subplot represent discrete-time observations. (Online version in colour.)

where $X_t \in \mathbb{R}^d$, $\theta \in \Theta$ is a parameter, $X_0 = x_\star$ is the initial condition with $x_\star$ given, $a : \Theta \times \mathbb{R}^d \to \mathbb{R}^d$ denotes the drift term, $\sigma : \mathbb{R}^d \to \mathbb{R}^{d \times d}$ denotes the diffusion coefficient and $\{W_t\}_{t \geq 0}$ is a standard $d$-dimensional Brownian motion. In practice, it is often difficult to have direct access to such continuous processes, where instead one has discrete-time partial observations of the process $\{X_t\}_{t \geq 0}$, denoted as $Y_{t_1}, \ldots, Y_{t_n}$, where $0 < t_1 < \ldots < t_n = T$, such that $Y_{t_p} \in \mathbb{R}^{d_y}$. Such processes are referred to as partially observed diffusion processes (PODPs), where one is interested in doing inference on the hidden process (1.1) given the observations. In order to do such inference, one must time-discretize such a process which induces a discretization bias. For (1.1), this can arise through common discretization forms such as an Euler or Milstein scheme [6]. Therefore, an important question, related to inference, is how one can reduce, or remove the discretization bias. Such a discussion motivates the development and implementation of unbiased estimators.

## (a) Motivating example

To help motivate unbiased estimation for PODPs, we provide an interesting application, for which we will test in this work. Our model example is the Fitzhugh–Nagumo (FHN) model for a neuron, which is a second-order ODE model arising in neuroscience, describing the actional potential generation within a neuronal axon. We consider a stochastic version of it, which is represented as the following:

$$\begin{bmatrix} dX_t^{(1)} \\ dX_t^{(2)} \end{bmatrix} = \begin{bmatrix} \theta_1(X_t^{(1)} - (X_t^{(1)})^3 - X_t^{(2)}) \\ \theta_2 X_t^{(1)} - X_t^{(2)} + \theta_3 \end{bmatrix} dt + \begin{bmatrix} \sigma_1 \\ \sigma_2 \end{bmatrix} dW_t, \quad X_0 = u_0, \tag{1.2}$$

where $X_t$ is the membrane potential. There has been some interest in parameter estimation [7–9], related to various variants of the FHN model. This particular model is well known within the community, and commonly acts as a toy problem in the field of mathematical neurosciences. For this reason, we will use this example within our numerical experiments. In figure 1, we

provide a simulation of (1.2), for arbitrary choices of $\Theta = (\theta_1, \theta_2, \theta_3)$, which demonstrates the interesting behaviour and dynamics. In the plot, we have also plotted non-noisy observations for $X_t^{(1)}$.

## (b) Methodology

The unbiased estimation of PODPs has been an important, yet challenging topic. Some original seminal work on this has been the idea of exact simulation, proposed in various works [10–12]. The underlying idea behind exact simulation is that, through a particular transformation, one can acquire an unbiased estimator, subject to certain restrictions on the form of the diffusion and its dimension. Since then there have been a number of extensions aimed at going beyond this, w.r.t. to more general multidimensional diffusions and continuous-time dynamics [13,14]. However, attention has recently been paid to unbiased estimation, for Bayesian computation, through the work of Rhee and Glynn [15,16], in which they provide unbiased and finite variance estimators through introducing randomization. In particular, these methods allow us to unbiasedly estimate an expectation of a functional, by randomizing on the level of the time-discretization in a type of multilevel Monte Carlo (MLMC) approach [17], where there is a coupling between different levels. As a result, this methodology has been considered in the context of both filtering and Bayesian computation [18–21] and gradient estimation [22]. One advantage of this approach is that, with couplings, it is relatively simple to use & implement computationally, while exploiting such methodologies on a range of different model problems or set-ups.

In this work, we are interested in developing an unbiased estimator of the Hessian for PODPs. Typically, the Hessian is not required for PODPS but currently this is of interest as current state-of-the-art stochastic gradient methodologies exploit Hessian information for an improved speed of convergence. Methods using this information include Newton type methods [23,24], which have improved rates of convergence over first-order stochastic gradient methods. It is also well known that these methods are typically biased as one does not fully require the whole Hessian, due to the computational burden. Therefore, this provides our motivation for firstly developing an unbiased estimator, and secondly for the Hessian. In order to develop an unbiased estimator, our methodology will largely follow that described in [22], with the extension of this from the score function to the Hessian. Other works that consider unbiased estimation of the gradient include [25,26]. In particular, we will exploit the use of the conditional particle filter (CPF), first considered by Andrieu *et al.* [27,28]. We provide an expression for the Hessian of the likelihood, while introducing an Euler time-discretization of the diffusion process in order to implement our unbiased estimator. We then describe how one can attain unbiased estimators, which is based on various couplings of the CPF. From this, we test this methodology to that of using the methods of [18,29] for the Hessian computation, as for a comparison, where we demonstrate the unbiased estimator through both the variance and bias. This will be conducted on both a single and multidimensional Ornstein–Uhlenbeck (OU) process, as well as a more complicated form of the FHN model. We remark that our estimator of the hessian is unbiased, but if the inverse hessian is required, it is possible to adapt the forthcoming methodology to that context as well.

## (c) Outline

In §2, we present our setting for our diffusion process. We also present a derived expression for the Hessian, with an appropriate time-discretization. Then, in §3, we describe our algorithm in detail for the unbiased estimator of the Hessian. This will be primarily based on a coupling of a coupled CPF. This will lead to §4 where we present our numerical experiments, which provide variance and bias plots. We compare the methodology of this work with that of the Delta particle filter. This comparison will be tested on a range of diffusion processes, which include an OU process and the FHN model. We summarize our findings in §5.

# 2. Model

In this section, we introduce our setting and notation regarding our partially observed diffusions. This will include a number of assumptions. We will then provide an expression for the Hessian of the likelihood function, with a time-discretization based on the Euler scheme. This will include a discussion on the stochastic model where we define the marginal likelihood. Finally, we present a result indicating the approximation of the Hessian computation as we take the limit of the discretization level.

## (a) Notation

Let $(X, \mathcal{X})$ be a measurable space. For $\varphi : X \to \mathbb{R}$, we write $\mathcal{B}_b(X)$ as the collection of bounded measurable functions, $\mathcal{C}^j(X)$ are the collection of $j$-times, $j \in \mathbb{N}$ continuously differentiable functions and we omit the subscript $j$ if the functions are simply continuous; if $\varphi : X \to \mathbb{R}^d$ we write $\mathcal{C}^j_d(X)$ and $\mathcal{C}_d(X)$. Let $\varphi : \mathbb{R}^d \to \mathbb{R}$, $\mathrm{Lip}_{||\cdot||_2}(\mathbb{R}^d)$ denote the collection of real-valued functions that are Lipschitz w.r.t. $||\cdot||_2$ ($||\cdot||_p$ denotes the $\mathbb{L}_p$−norm of a vector $x \in \mathbb{R}^d$). That is, $\varphi \in \mathrm{Lip}_{||\cdot||_2}(\mathbb{R}^d)$ if there exists a $C < +\infty$ such that for any $(x, y) \in \mathbb{R}^{2d}$

$$|\varphi(x) - \varphi(y)| \leq C||x - y||_2.$$

We write $||\varphi||_{\mathrm{Lip}}$ as the Lipschitz constant of a function $\varphi \in \mathrm{Lip}_{||\cdot||_2}(\mathbb{R}^d)$. For $\varphi \in \mathcal{B}_b(X)$, we write the supremum norm $||\varphi|| = \sup_{x \in X} |\varphi(x)|$. $\mathcal{P}(X)$ denotes the collection of probability measures on $(X, \mathcal{X})$. For a measure $\mu$ on $(X, \mathcal{X})$ and a $\varphi \in \mathcal{B}_b(X)$, the notation $\mu(\varphi) = \int_X \varphi(x)\mu(dx)$ is used. $B(\mathbb{R}^d)$ denote the Borel sets on $\mathbb{R}^d$. $dx$ is used to denote the Lebesgue measure. Let $K : X \times \mathcal{X} \to [0, \infty)$ be a non-negative operator and $\mu$ be a measure, then we use the notations $\mu K(dy) = \int_X \mu(dx)K(x, dy)$ and for $\varphi \in \mathcal{B}_b(X)$, $K(\varphi)(x) = \int_X \varphi(y)K(x, dy)$. For $A \in \mathcal{X}$, the indicator is written $\mathbb{I}_A(x)$. $\mathcal{U}_A$ denotes the uniform distribution on the set $A$. $\mathcal{N}_s(\mu, \Sigma)$ (resp. $\psi_s(x; \mu, \Sigma)$) denotes an $s$-dimensional Gaussian distribution (density evaluated at $x \in \mathbb{R}^s$) of mean $\mu$ and covariance $\Sigma$. If $s = 1$ we omit the subscript $s$. For a vector/matrix $X$, $X^*$ is used to denote the transpose of $X$. For $A \in \mathcal{X}$, $\delta_A(du)$ denotes the Dirac measure of $A$, and if $A = \{x\}$ with $x \in X$, we write $\delta_x(du)$. For a vector-valued function in $d$-dimensions (resp. $d$-dimensional vector), $\varphi(x)$ (resp. $x$) say, we write the $i$th−component ($i \in \{1, \dots, d\}$) as $\varphi(x)^{(i)}$ (resp. $x^{(i)}$). For a $d \times q$ matrix $x$, we write the $(i, j)^{\mathrm{th}}$−entry as $x^{(ij)}$. For $\mu \in \mathcal{P}(X)$ and $X$ a random variable on $X$ with distribution associated with $\mu$ we use the notation $X \sim \mu(\cdot)$.

## (b) Diffusion process

Let $\theta \in \Theta \subseteq \mathbb{R}^{d_\theta}$ be fixed and we consider a diffusion process on the probability space $(\Omega, \mathcal{F}, \{\mathcal{F}\}_{t \geq 0}, \mathbb{P}_\theta)$, such that

$$dX_t = a_\theta(X_t) \, dt + \sigma(X_t) \, dW_t, \quad X_0 = x_\star \in \mathbb{R}^d, \tag{2.1}$$

where $X_t \in \mathbb{R}^d$, $X_0 = x_\star$ with $x_\star$ given, $a : \Theta \times \mathbb{R}^d \to \mathbb{R}^d$ is the drift term, $\sigma : \mathbb{R}^d \to \mathbb{R}^{d \times d}$ is the diffusion coefficient and $\{W_t\}_{t \geq 0}$ is a standard $d$-dimensional Brownian motion. We assume that for any fixed $\theta \in \Theta$, $a^{(i)}_\theta \in \mathcal{C}^2(\mathbb{R}^d)$ and $\sigma^{(ij)} \in \mathcal{C}^2(\mathbb{R}^d)$ for $(i, j) \in \{1, \dots, d\}^2$. For fixed $x \in \mathbb{R}^d$, we have $a_\theta(x)^{(i)} \in \mathcal{C}(\Theta)$ for $i \in \{1, \dots, d\}$.

Furthermore, we make the following additional assumption, termed (D1).

  (i) *Uniform ellipticity*: $\Sigma(x) := \sigma(x)\sigma(x)^*$ is uniformly positive definite over $x \in \mathbb{R}^d$.
  (ii) *Globally Lipschitz*: for any $\theta \in \Theta$, there exists a positive constant $C < \infty$ such that

$$|a_\theta(x)^{(i)} - a_\theta(x')^{(i)}| + |\sigma(x)^{(ij)} - \sigma(x')^{(ij)}| \leq C||x - x'||_2,$$

for all $(x, x') \in \mathbb{R}^d \times \mathbb{R}^d$, $(i, j) \in \{1, \dots, d\}^2$.

Let $0 < t_1, < \ldots 0$ be a given collection of time points. Following [22], by the use of Girsanov Theorem, for any $\mathbb{P}_\theta$-integrable $\varphi : \Theta \times \mathbb{R}^{nd} \to \mathbb{R}$,

$$\mathbb{E}_\theta[\varphi_\theta(X_{t_1}, \ldots, X_{t_n})] = \mathbb{E}_\mathbb{Q}\left[\varphi_\theta(X_{t_1}, \ldots, X_{t_n}) \frac{d\mathbb{P}_\theta}{d\mathbb{Q}}(\mathbf{X}_T)\right], \tag{2.2}$$

where $\mathbb{E}_\theta$ denotes the expectation w.r.t. $\mathbb{P}_\theta$, set $\mathbf{X}_T = \{X_t\}_{t \in [0,T]}$, and the change of measure is given by

$$\frac{d\mathbb{P}_\theta}{d\mathbb{Q}}(\mathbf{X}_T) = \exp\left\{-\frac{1}{2}\int_0^T \|b_\theta(X_s)\|_2^2 \, ds + \int_0^T b_\theta(X_s)^* \, dW_s\right\},$$

with $b_\theta(x) = \Sigma(x)^{-1}\sigma(x)^* a_\theta(x)$ is a $d$-dimensional vector. Below, we consider a change of measure to the law $\mathbb{Q}$, which is induced by using that $dX_t = \sigma(X_t) \, dW_t$, where $X_t$ solves such a process. Since $\mathbb{P}_\theta$ and $\mathbb{Q}$ are equibilant, therefore by Girsanov's Theorem

$$\rho_\theta(\mathbf{X}_T) = \varphi_\theta(X_{t_1:t_n}) \frac{d\mathbb{P}_\theta}{d\mathbb{Q}}(\mathbf{X}_T),$$

where the corresponding Radon–Nikodym derivative is

$$\frac{d\mathbb{P}_\theta}{d\mathbb{Q}}(\mathbf{X}_T) = \exp\left\{-\frac{1}{2}\int_0^T \|b_\theta(X_s)\|_2^2 \, ds + \int_0^T b_\theta(X_s)^* \Sigma(X_s)^- 1\sigma(X_s)^* \, dX_s\right\}.$$

Now if we assume that $\varphi_\theta$ is differentiable w.r.t. $\theta$, then one has for $i \in \{1, \ldots, d_\theta\}$

$$\mathfrak{G}_\theta^{(i)} := \frac{\partial}{\partial\theta^{(i)}}\left(\log\{\mathbb{E}_\theta[\varphi_\theta(X_{t_1}, \ldots, X_{t_n})]\}\right) = \mathbb{E}_{\overline{\mathbb{P}}_\theta}\left[\frac{\partial}{\partial\theta^{(i)}}\left(\log\{\rho_\theta(\mathbf{X}_T)\}\right)\right], \tag{2.3}$$

where $\overline{\mathbb{P}}_\theta = \varphi_\theta \mathbb{P}_\theta / \mathbb{P}_\theta(\varphi_\theta)$ and $\mathbb{P}_\theta[\varphi_\theta(X_{t_1}, \ldots, X_{t_n})]$ is the law of the diffusion process (1.1). From herein, we will use the short-hand notation $\varphi_\theta(X_{t_1}, \ldots, X_{t_n}) = \varphi_\theta(X_{t_1:t_n})$ and also set, for $i \in \{1, \ldots, d_\theta\}$,

$$G_\theta(\mathbf{X}_T)^{(i)} = \frac{\partial}{\partial\theta^{(i)}}\left(\log\{\rho_\theta(\mathbf{X}_T)\}\right).$$

## (c) Hessian expression

Given the expression (2.3) our objective is now to write the matrix of second derivatives, for $(i, j) \in \{1, \ldots, d_\theta\}^2$

$$\mathfrak{H}_\theta^{(ij)} := -\frac{\partial^2}{\partial\theta^{(i)}\partial\theta^{(j)}}\left(\log\{\mathbb{E}_\theta[\varphi_\theta(X_{t_1:t_n})]\}\right),$$

in terms of expectations w.r.t. $\overline{\mathbb{P}}_\theta$.

We have the following simple calculation:

$$\frac{\partial^2}{\partial\theta^{(i)}\partial\theta^{(j)}}\left(\log\{\mathbb{E}_\theta[\varphi_\theta(X_{t_1:t_n})]\}\right) = \frac{\partial}{\partial\theta^{(i)}}\left(\frac{(\partial/\partial\theta^{(j)})\{\mathbb{E}_\theta[\varphi_\theta(X_{t_1:t_n})]\}}{\mathbb{E}_\theta[\varphi_\theta(X_{t_1:t_n})]}\right)$$

$$= \frac{\frac{\partial^2}{\partial\theta^{(i)}\partial\theta^{(j)}}\{\mathbb{E}_\theta[\varphi_\theta(X_{t_1:t_n})]\}}{\mathbb{E}_\theta[\varphi_\theta(X_{t_1:t_n})]}$$

$$- \frac{\frac{\partial}{\partial\theta^{(i)}}\{\mathbb{E}_\theta[\varphi_\theta(X_{t_1:t_n})]\}(\partial/\partial\theta^{(j)})\{\mathbb{E}_\theta[\varphi_\theta(X_{t_1:t_n})]\}}{\mathbb{E}_\theta[\varphi_\theta(X_{t_1:t_n})]^2}$$

$$=: T_1 - T_2.$$

Under relatively weak conditions, one can express $T_1$ and $T_2$ as

$$T_1 = \mathbb{E}_{\overline{\mathbb{P}}_\theta}\left[\frac{\partial\log\{\rho_\theta(\mathbf{X}_T)\}}{\partial\theta^{(i)}}\frac{\partial\log\{\rho_\theta(\mathbf{X}_T)\}}{\partial\theta^{(j)}}\right] + \mathbb{E}_{\overline{\mathbb{P}}_\theta}\left[\frac{\partial^2}{\partial\theta^{(i)}\partial\theta^{(j)}}\left(\log\{\rho_\theta(\mathbf{X}_T)\}\right)\right]$$

and

$$T_2 = \mathbb{E}_{\overline{\mathbb{P}}_\theta}\left[\frac{\partial\log\{\rho_\theta(\mathbf{X}_T)\}}{\partial\theta^{(i)}}\right]\mathbb{E}_{\overline{\mathbb{P}}_\theta}\left[\frac{\partial\log\{\rho_\theta(\mathbf{X}_T)\}}{\partial\theta^{(j)}}\right].$$

Therefore, we have the following expression:

$$\mathfrak{H}_{\theta}^{(ij)} = \mathbb{E}_{\overline{\mathbb{P}}_{\theta}}\left[\frac{\partial \log\{\rho_{\theta}(\mathbf{X}_T)\}}{\partial \theta^{(i)}}\right] \mathbb{E}_{\overline{\mathbb{P}}_{\theta}}\left[\frac{\partial \log\{\rho_{\theta}(\mathbf{X}_T)\}}{\partial \theta^{(j)}}\right]$$

$$- \mathbb{E}_{\overline{\mathbb{P}}_{\theta}}\left[\frac{\partial \log\{\rho_{\theta}(\mathbf{X}_T)\}}{\partial \theta^{(i)}}\frac{\partial \log\{\rho_{\theta}(\mathbf{X}_T)\}}{\partial \theta^{(j)}}\right] - \mathbb{E}_{\overline{\mathbb{P}}_{\theta}}\left[\frac{\partial^2}{\partial \theta^{(i)}\partial \theta^{(j)}}\Big(\log\{\rho_{\theta}(\mathbf{X}_T)\}\Big)\right]. \quad (2.4)$$

Defining, for $(i,j) \in \{1,\ldots,d_{\theta}\}^2$

$$H_{\theta}(\mathbf{X}_T)^{(ij)} := \frac{\partial^2}{\partial \theta^{(i)}\partial \theta^{(j)}}\Big(\log\{\rho_{\theta}(\mathbf{X}_T)\}\Big),$$

one can write more succinctly

$$\mathfrak{H}_{\theta}^{(ij)} = \mathbb{E}_{\overline{\mathbb{P}}_{\theta}}[G_{\theta}(\mathbf{X}_T)^{(i)}]\mathbb{E}_{\overline{\mathbb{P}}_{\theta}}[G_{\theta}(\mathbf{X}_T)^{(j)}] - \mathbb{E}_{\overline{\mathbb{P}}_{\theta}}[G_{\theta}(\mathbf{X}_T)^{(i)}G_{\theta}(\mathbf{X}_T)^{(j)}] - \mathbb{E}_{\overline{\mathbb{P}}_{\theta}}[H_{\theta}(\mathbf{X}_T)^{(ij)}].$$

## (d) Stochastic model

Consider a sequence of random variables $(Y_{t_1},\ldots,Y_{t_n})$ where $0 < t_1 < \ldots < t_n = T$, where $Y_{t_p} \in \mathbb{R}^{d_y}$, which are assumed to have the following joint conditional Lebesgue density

$$p_{\theta}(y_{t_1},\ldots,y_{t_n}|\{x_s\}_{0\leq s\leq T}) = \prod_{p=1}^{n} g_{\theta}(y_{t_p}|x_{t_p}),$$

where $g: \Theta \times \mathbb{R}^d \times \mathbb{R}^{d_y} \to \mathbb{R}^+$ for any $(\theta,x) \in \Theta \times \mathbb{R}^d$, $\int_{\mathbb{R}^{d_y}} g_{\theta}(y|x)dy = 1$ such that $dy$ is the Lebesgue measure. Now if one considers instead realizations of the random variables $(Y_{t_1},\ldots,Y_{t_n})$, in the conditioning of the joint density we have a state-space model with marginal likelihood

$$p_{\theta}(y_{t_1},\ldots,y_{t_n}) := \mathbb{E}_{\theta}\left[\prod_{p=1}^{n} g_{\theta}(y_{t_p}|X_{t_p})\right].$$

Note that the framework to be investigated in this article is not restricted to this special case, but we shall focus on it for the rest of the paper. So to clarify $\varphi_{\theta}(x_{t_1},\ldots,x_{t_n}) = \prod_{p=1}^{n} g_{\theta}(y_{t_p}|x_{t_p})$ from herein.

In reference to (2.3) and (2.4), we have that

$$\frac{\partial \log\{\rho_{\theta}(\mathbf{X}_T)\}}{\partial \theta^{(i)}} = \sum_{p=1}^{n} \frac{\partial}{\partial \theta^{(i)}}\Big(\log\{g_{\theta}(y_{t_p}|x_{t_p})\}\Big) - \frac{1}{2}\int_0^T \frac{\partial}{\partial \theta^{(i)}}\Big(||b_{\theta}(X_s)||_2^2\Big)ds$$

$$+ \frac{\partial}{\partial \theta^{(i)}}\left(\int_0^T b_{\theta}(X_s)^*\Sigma(X_s)^{-1}\sigma(X_s)^* \, dX_s\right)$$

and

$$\frac{\partial^2}{\partial \theta^{(i)}\partial \theta^{(j)}}\Big(\log\{\rho_{\theta}(\mathbf{X}_T)\}\Big) = \sum_{p=1}^{n} \frac{\partial^2}{\partial \theta^{(i)}\partial \theta^{(j)}}\Big(\log\{g_{\theta}(y_{t_p}|x_{t_p})\}\Big)$$

$$- \frac{1}{2}\int_0^T \frac{\partial^2}{\partial \theta^{(i)}\partial \theta^{(j)}}\Big(||b_{\theta}(X_s)||_2^2\Big)ds + \frac{\partial^2}{\partial \theta^{(i)}\partial \theta^{(j)}}\left(\int_0^T b_{\theta}(X_s)^*\Sigma(X_s)^{-1}\sigma(X_s)^* \, dX_s\right).$$

## (e) Time-discretization

From herein, we take the simplification that $t_p = p, p \in \{1,\ldots,n\}, T = n$. Let $l \in \mathbb{N}_0$ be a given level of discretization, and consider the Euler discretization of step size $\Delta_l = 2^{-l}, k \in \{1,2,\ldots,\Delta_l^{-1}T\}$

with $\tilde{X}_0 = x_\star$:

$$\tilde{X}_{k\Delta_l} = \tilde{X}_{(k-1)\Delta_l} + a_\theta(\tilde{X}_{(k-1)\Delta_l})\Delta_l + \sigma(\tilde{X}_{(k-1)\Delta_l})[W_{k\Delta l} - W_{(k-1)\Delta_l}]. \tag{2.5}$$

Set $\mathbf{x}_T^l = (x_\star, \tilde{x}_{\Delta_l}, \ldots, \tilde{x}_T)$. We then consider the vector-valued function $G^l : \Theta \times (\mathbb{R}^d)^{\Delta_l^{-1}T+1} \to \mathbb{R}^{d_\theta}$ and the matrix-valued function $H^l : \Theta \times (\mathbb{R}^d)^{\Delta_l^{-1}T+1} \to \mathbb{R}^{d_\theta \times d_\theta}$ defined as, for $(i, j) \in \{1, \ldots, d_\theta\}^2$

$$G_\theta^l(\mathbf{x}_T^l)^{(i)} = \sum_{p=1}^n \frac{\partial}{\partial\theta^{(i)}}\left(\log\{g_\theta(y_p|\tilde{x}_p)\}\right) - \frac{\Delta_l}{2}\sum_{k=0}^{\Delta_l^{-1}T-1}\frac{\partial}{\partial\theta^{(i)}}\left(||b_\theta(\tilde{x}_{k\Delta_l})||_2^2\right)$$

$$+ \sum_{k=0}^{\Delta_l^{-1}T-1}\frac{\partial}{\partial\theta^{(i)}}\left(b_\theta(\tilde{x}_{k\Delta_l})^*\Sigma(\tilde{x}_{k\Delta_l})^{-1}\sigma(\tilde{x}_{k\Delta_l})^*[\tilde{x}_{(k+1)\Delta_l} - \tilde{x}_{k\Delta_l}]\right)$$

and

$$H_\theta^l(\mathbf{x}_T^l)^{(ij)} = \sum_{p=1}^n \frac{\partial^2}{\partial\theta^{(i)}\partial\theta^{(j)}}\left(\log\{g_\theta(y_p|\tilde{x}_p)\}\right) - \frac{\Delta_l}{2}\sum_{k=0}^{\Delta_l^{-1}T-1}\frac{\partial^2}{\partial\theta^{(i)}\partial\theta^{(j)}}\left(||b_\theta(\tilde{x}_{k\Delta_l})||_2^2\right)$$

$$+ \sum_{k=0}^{\Delta_l^{-1}T-1}\frac{\partial^2}{\partial\theta^{(i)}\partial\theta^{(j)}}\left(b_\theta(\tilde{x}_{k\Delta_l})^*\Sigma(\tilde{x}_{k\Delta_l})^{-1}\sigma(\tilde{x}_{k\Delta_l})^*[\tilde{x}_{(k+1)\Delta_l} - \tilde{x}_{k\Delta_l}]\right).$$

Then, noting (2.4), we have an Euler approximation of the Hessian

$$\mathfrak{H}_\theta^{l,(ij)} := \frac{\mathbb{E}_\theta[\varphi_\theta(\tilde{X}_{1:n})G_\theta^l(\mathbf{X}_T^l)^{(i)}]}{\mathbb{E}_\theta[\varphi_\theta(\tilde{X}_{1:n})]}\frac{\mathbb{E}_\theta[\varphi_\theta(\tilde{X}_{1:n})G_\theta^l(\mathbf{X}_T^l)^{(j)}]}{\mathbb{E}_\theta[\varphi_\theta(\tilde{X}_{1:n})]}$$
$$- \frac{\mathbb{E}_\theta[\varphi_\theta(\tilde{X}_{1:n})G_\theta^l(\mathbf{X}_T^l)^{(i)}G_\theta^l(\mathbf{X}_T^l)^{(j)}]}{\mathbb{E}_\theta[\varphi_\theta(\tilde{X}_{1:n})]} - \frac{\mathbb{E}_\theta[\varphi_\theta(\tilde{X}_{1:n})H_\theta^l(\mathbf{X}_T^l)^{(ij)}]}{\mathbb{E}_\theta[\varphi_\theta(\tilde{X}_{1:n})]}.$$

In the context of the model in §d, if one sets

$$\pi_\theta^l(d\mathbf{x}_T^l) \propto \left\{\prod_{p=1}^n g_\theta(y_p|\tilde{x}_p)p_\theta^l(\tilde{x}_{p-1}|\tilde{x}_p)\right\}d\mathbf{x}_T^l,$$

where $p_\theta^l$ is the transition density induced by discretized diffusion process (2.5) (over unit time), and we use the abuse of notation that $d\mathbf{x}_T^l$ is the Lebesgue measure on the coordinates $(\tilde{x}_{\Delta_l}, \ldots, \tilde{x}_T)$, then one has that

$$\mathfrak{H}_\theta^{l,(ij)} = \pi_\theta^l(G_\theta^{l,(i)})\pi_\theta^l(G_\theta^{l,(j)}) - \pi_\theta^l(G_\theta^{l,(i)}G_\theta^{l,(j)}) - \pi_\theta^l(H_\theta^{l,(ij)}), \tag{2.6}$$

where we are using the short-hand $G_\theta^{l,(i)} = G_\theta^l(\mathbf{X}_T^l)^{(i)}$ and $H_\theta^{l,(ij)} = H_\theta^l(\mathbf{X}_T^l)^{(ij)}$ etc.

We have the following result whose proof and assumption (D2) is in appendix A.

**Proposition 2.1.** *Assume (D1-D2). Then for any $(i, j) \in \{1, \ldots, d_\theta\}^2$, we have*

$$\lim_{l\to\infty}\mathfrak{H}_\theta^{l,(ij)} = \mathfrak{H}_\theta^{(ij)}.$$

The main strategy of the proof is by strong convergence, which means that one can characterize an upper-bound on $|\mathfrak{H}_\theta^{l,(ij)} - \mathfrak{H}_\theta^{(ij)}|$ of $\mathcal{O}(\Delta_l^{1/2})$ but that rate is most likely not sharp, as one expects $\mathcal{O}(\Delta_l)$.

## 3. Algorithm

The objective of this section, using only approximations of (2.6), is to obtain an unbiased estimate of $\mathfrak{H}_\theta^{(ij)}$ for any fixed $\theta \in \Theta$ and $(i, j) \in \{1, \ldots, d_\theta\}^2$. Our approach is essentially an application of the methodology in [22] and so we provide a review of that approach in the sequel.

## (a) Strategy

To focus our description, we shall suppose that we are interested in computing an unbiased estimate of $\mathfrak{G}_\theta^{(i)}$ for some fixed $i$; we remark that this specialization is not needed and is only used for notational convenience. An Euler approximation of $\mathfrak{G}_\theta^{(i)}$ is $\pi_\theta^l(G_\theta^{l,(i)}) =: \mathfrak{G}_\theta^{l,(i)}$. To further simplify the notation, we will simply write $G_\theta^l$ instead of $G_\theta^{l,(i)}$.

Suppose that one can construct a sequence of random variables $(\hat{\pi}_\theta^l(G_\theta^l))_{l \in \mathbb{N}_0}$ on a potentially extended probability space with expectation operator $\overline{\mathbb{E}}_\theta$, such that for each $l \in \mathbb{N}_0$, $\overline{\mathbb{E}}_\theta[\hat{\pi}_\theta^l(G_\theta^l)] = \pi_\theta^l(G_\theta^l)$. Moreover, consider the independent sequence of random variables, $(\Xi_\theta^l)_{l \in \mathbb{N}_0}$ which are constructed so that for $l \in \mathbb{N}_0$

$$\overline{\mathbb{E}}_\theta[\Xi_\theta^l] := \overline{\mathbb{E}}_\theta[\hat{\pi}_\theta^l(G_\theta^l)] - \overline{\mathbb{E}}_\theta[\hat{\pi}_\theta^{l-1}(G_\theta^{l-1})] = \pi_\theta^l(G_\theta^l) - \pi_\theta^{l-1}(G_\theta^{l-1}), \tag{3.1}$$

with $\overline{\mathbb{E}}_\theta[\hat{\pi}_\theta^{-1}(G_\theta^{-1})] := \pi_\theta^{-1}(G_\theta^{-1}) := 0$. Now let $\mathbb{P}_L$ be a positive probability mass function on $\mathbb{N}_0$ and set $\overline{\mathbb{P}}_L(l) = \sum_{p=l}^{\infty} \mathbb{P}_L(p)$. Now if,

$$\sum_{l \in \mathbb{N}_0} \frac{1}{\overline{\mathbb{P}}_L(l)} \left\{ \overline{\mathrm{Var}}_\theta[\Xi_\theta^l] + (\mathfrak{G}_\theta^{l,(i)} - \mathfrak{G}_\theta^{(i)})^2 \right\} < +\infty, \tag{3.2}$$

then if one samples $L$ from $\mathbb{P}_L$ independently of the sequence $(\Xi_l)_{l \in \mathbb{N}_0}$ then by e.g. ([17], Theorem 5) the estimate

$$\widehat{\mathfrak{G}}_\theta^{(i)} := \sum_{l=0}^{L} \frac{\Xi_\theta^l}{\overline{\mathbb{P}}_L(l)}, \tag{3.3}$$

is an unbiased and finite variance estimator of $\mathfrak{G}_\theta^{(i)}$. Before describing in fuller detail our approach, which requires numerous techniques and methodologies, we first present our main result which is an unbiased theorem related to our estimator of the Hessian. This is given through the following proposition.

**Proposition 3.1.** *Assume (D1-D2). Then there exists choices of $\mathbb{P}_L$ so that* (3.9) *is an unbiased and finite variance estimator of $\mathfrak{H}_\theta^{(ij)}$ for each $(i, j) \in D_\theta$.*

*Proof.* This is the same as ([22], theorem 2), except one must repeat the arguments of that paper given lemma A.3 in the appendix and given the rate in the proof of proposition 2.1. Since the arguments and calculations are almost identical, they are omitted in their entirety. ∎

The main point is that the choice of $\mathbb{P}_L$ is as in [22], which is: in the case that $\sigma$ is constant $\mathbb{P}_L(l) \propto \Delta_l(l+1)\log_2(2+l)^2$ and in the non-constant case $\mathbb{P}_L(l) \propto \Delta_l^{1/2}(l+1)\log_2(2+l)^2$; both choices achieve finite variance and costs to achieve an error of $\mathcal{O}(\epsilon)$ with high probability as in ([16], propositions 4 and 5).

The main issue is to construct the sequence of independent random variables $(\Xi_\theta^l)_{l \in \mathbb{N}_0}$ such that (3.1) and (3.2) hold and that the expected computational cost for doing so is not unreasonable as a functional of $\Delta_l$: a method for doing this is in [22] as we will now describe.

## (b) Computing $\Xi_\theta^0$

The computation of $\Xi_\theta^0$ is performed by using exactly the coupled conditional particle filter (CCPF) that has been introduced in [30]. This is an algorithm which allows one to construct a random variable $\hat{\pi}_\theta^0(G_\theta^0)$ such that $\overline{\mathbb{E}}_\theta[\hat{\pi}_\theta^0(G_\theta^0)] = \pi_\theta^0(G_\theta^0)$ and we will set $\Xi_\theta^0 = \hat{\pi}_\theta^0(G_\theta^0)$.

We begin by introducing the Markov kernel $C^l : \Theta \times \mathsf{X}^l \to \mathcal{P}(\mathsf{X}^l)$ in algorithm 1. To that end, we will use the notation $x_{\Delta_l:k}^{i,l} \in (\mathbb{R}^d)^{k\Delta_l^{-1}}$, where $l \in \mathbb{N}_0$ is the level of discretization, $i \in \{1, \ldots, N\}$ is a particle (sample) indicator, $k \in \{1, \ldots, n\}$ is a time parameter and $x_{\Delta_l:k}^{i,l} = (x_{\Delta_l}^{i,l}, x_{2\Delta_l}^{i,l}, \ldots, x_k^{i,l})$. The kernel described in algorithm 1 is called the called the CPF, as developed in [27], and allows one to generate, under minor conditions, an ergodic Markov chain of invariant measure $\pi_\theta^l$. By itself,

**Algorithm 1.** Conditional Particle Filter at level $l \in \mathbb{N}_0$.

1. Input $x_{\Delta_l:n}' \in \mathsf{X}^l$. Set $k=1$, $x_0^i = x_\star$, $a_0^i = i$ for $i \in \{1,\ldots,N-1\}$.

2. Sampling: for $i \in \{1,\ldots,N-1\}$ sample $x_{k-1+\Delta_l:k}^i | x_{k-1}^{a_{k-1}^i}$ using the Markov kernel $p_\theta^l$. Set

   $x_{k-1+\Delta_l:k}^N = x_k'$ and for $i \in \{1,\ldots,N-1\}$, $x_{\Delta_l:k}^i = (x_{\Delta_l:k-1}^{a_{k-1}^i}, x_{k-1+\Delta_l:k}^i)$. If $k=n$ go to 4.

3. Resampling: Construct the probability mass function on $\{1,\ldots,N\}$:

$$r_1^i = \frac{g_\theta(y_k|x_k^i)}{\sum_{j=1}^N g_\theta(y_k|x_k^j)}.$$

   For $i \in \{1,\ldots,N-1\}$ sample $a_k^i$ from $r_1^i$. Set $k=k+1$ and return to the start of 2.

4. Construct the probability mass function on $\{1,\ldots,N\}$:

$$r_1^i = \frac{g_\theta(y_n|x_n^i)}{\sum_{j=1}^N g_\theta(y_n|x_n^j)}.$$

   Sample $i \in \{1,\ldots,N\}$ using this mass function and return $x_{\Delta_l:n}^i$.

it does not provide unbiased estimates of expectations w.r.t. $\pi_\theta^l$, unless $\pi_\theta^l$ is the initial distribution of the Markov chain. However, the kernel will be of use in our subsequent discussion.

Our approach generates a Markov chain $\{Z_m\}_{m\in\mathbb{N}_0}$ on the space $\mathsf{Z}^0 := \mathsf{X}^0 \times \mathsf{X}^0$, $Z_m \in \mathsf{Z}^0$. In order to describe how one can simulate this Markov chain, we introduce several objects which will be needed. The first of these is the kernel $\check{p}^l : \Theta \times \mathbb{R}^{2d} \to \mathcal{P}(\mathbb{R}^{2d})$, which we need in the case $l=0$ and its simulation is described in algorithm 2. The Markov kernel is used to simulate intermediate points from $x_{k-1}$ to the next observations at time $k$, with time step $\Delta_l$. We will also need to simulate the maximal coupling of two probability mass functions on $\{1,\ldots,N\}$, for some $N \in \mathbb{N}$, and this is described in algorithm 3.

**Remark 3.2.** Step 4 of algorithm 3 can be modified to the case where one generates the pair $(i,j) \in \{1,\ldots,N\}^2$ from any coupling of the two probability mass functions $(r_4,r_5)$. In our simulations in §4, we will do this by sampling by inversion from $(r_4,r_5)$, using the same uniform random variable. However, to simplify the mathematical analysis that we will give in the appendix, we consider exactly algorithm 3 in our calculations.

To describe the CCPF kernel, we must first introduce a driving CCPF, which is presented in algorithm 4. The driving CCPF is nothing more than an ordinary coupled particle filter, except the final pair of trajectories is 'frozen' as is given in the algorithm (that is $(x_{1:n}', \bar{x}_{1:n}')$ as in step 1 of algorithm 4) and allowed to interact with the rest of the particle system. Given the ingredients in algorithms 2–4, we are now in a position to describe the CCPF kernel, which is a Markov kernel $K^0 : \Theta \times \mathsf{Z}^0 \to \mathcal{P}(\mathsf{Z}^0)$, whose simulation is presented in algorithm 5. We will consider the Markov chain $\{Z_m\}_{m\in\mathbb{N}_0}$, $Z_m = (X_{1:n}(m), \bar{X}_{1:n}(m))$, generated by the CCPF kernel in algorithm 5 and with initial distribution

$$\nu_\theta^0\Big(d(x_{1:n}, \bar{x}_{1:n})\Big)$$

$$= \int_{\mathsf{Z}^0} \Big(\prod_{k=1}^n p_\theta^0(x_{k-1}', x_k')\Big)\Big(\prod_{k=1}^n p_\theta^0(\bar{x}_{k-1}', \bar{x}_k')\Big) C_\theta^0(x_{1:n}', dx_{1:n})\delta_{\{\bar{x}_{1:n}'\}}(d\bar{x}_{1:n}), d(x_{1:n}, \bar{x}_{1:n}), \qquad (3.4)$$

where $x_0' = \bar{x}_0' = x_\star$.

We remark that in algorithm 5, marginally, $x_{1:n}^i$ (resp. $\bar{x}_{1:n}^j$) has been generated according to $C_\theta^0(x_{1:n}, \cdot)$ (resp. $C_\theta^0(\bar{x}_{1:n}, \cdot)$). A rather important point is that if the two input trajectories in step 1 of algorithm 5 are equal, i.e. $x_{1:n} = \bar{x}_{1:n}$, then the output trajectories will also be equal. To that end,

define the stopping time associated with the given Markov chain

$$\tau^0 = \inf\{m \geq 1 : x_{1:n}(m) = \bar{x}_{1:n}(m)\}.$$

Then, setting $m^* \in \{2, 3, \ldots\}$ one has the following estimator:

$$\hat{\pi}_\theta^0(G_\theta^0) := G_\theta^0(X_{1:n}(m^*)) + \sum_{m=m^*+1}^{\tau^0-1} \{G_\theta^0(X_{1:n}(m)) - G_\theta^0(\bar{X}_{1:n}(m))\}, \tag{3.5}$$

and one sets $\Xi_\theta^0 = \hat{\pi}_\theta^0(G_\theta^0)$. The procedure for computing $\Xi_\theta^0$ is summarized in algorithm 6.

---

**Algorithm 2**. Simulating the Kernel $\check{p}_\theta^l$.

1. Input $(x_0, \bar{x}_0) \in \mathbb{R}^{2d}$ and the level $l \in \mathbb{N}_0$.
2. Generate $V_{k\Delta_l} \overset{\text{i.i.d.}}{\sim} \mathcal{N}_d(0, \Delta_l I_d)$, for $k \in \{1, \ldots, \Delta_l^{-1}\}$.
3. Run the two recursions, for $k \in \{1, \ldots, \Delta_l^{-1}\}$:

$$\& X_{k\Delta_l} = X_{(k-1)\Delta_l} + a_\theta(X_{(k-1)\Delta_l})\Delta_l + \sigma(X_{(k-1)\Delta_l})V_{k\Delta_l}$$

$$\& \bar{X}_{k\Delta_l} = \bar{X}_{(k-1)\Delta_l} + a_\theta(\bar{X}_{(k-1)\Delta_l})\Delta_l + \sigma(\bar{X}_{(k-1)\Delta_l})V_{k\Delta_l}.$$

4. Return $(x_1, \bar{x}_1) \in \mathbb{R}^{2d}$.

---

**Algorithm 3**. Simulating a Maximal Coupling of Two Probability Mass Functions on $\{1, \ldots, N\}$.

1. Input: Two probability mass functions (PMFs) $(r_1^1, \ldots, r_1^N)$ and $(r_2^1, \ldots, r_2^N)$ on $\{1, \ldots, N\}$.
2. Generate $U \sim \mathcal{U}_{[0,1]}$.
3. If $U < \sum_{i=1}^N \min\{r_1^i, r_2^i\} =: \bar{r}$ then generate $i \in \{1, \ldots, N\}$ according to the probability mass function:

$$r_3^i = \frac{1}{\bar{r}}\min\{r_1^i, r_2^i\}$$

   and set $j = i$.
4. Otherwise generate $i \in \{1, \ldots, N\}$ and $j \in \{1, \ldots, N\}$ independently according to the probability mass functions

$$r_4^i = \frac{1}{1 - \bar{r}}(r_1^i - \min\{r_1^i, r_2^i\})$$

   and

$$r_5^j = \frac{1}{1 - \bar{r}}(r_2^j - \min\{r_1^j, r_2^j\}),$$

   respectively.
5. Output: $(i, j) \in \{1, \ldots, N\}^2$. $i$, marginally has PMF $r_1^i$ and $j$, marginally has PMF $r_2^j$.

---

## (c) Computing $(\Xi_\theta^l)_{l \in \mathbb{N}}$

We are now concerned with the task of computing $(\Xi_\theta^l)_{l \in \mathbb{N}}$ such that (3.1)–(3.2) are satisfied. Throughout the section $l \in \mathbb{N}$ is fixed. We will generate a Markov chain $\{\check{Z}_m^l\}_{m \in \mathbb{N}_0}$ on the space $Z^l \times Z^{l-1}$, where $Z^l = X^l \times X^l$ and $\check{Z}_m^l \in Z^l \times Z^{l-1}$. In order to construct our Markov chain kernel, as in the previous section, we will need to provide some algorithms. We begin with the Markov kernel $\check{q}^l : \Theta \times \mathbb{R}^{4d} \to \mathcal{P}(\mathbb{R}^{\Delta_l^{-1}2d} \times \mathbb{R}^{\Delta_{l-1}^{-1}2d})$ which will be needed and whose simulation is described in algorithm 7. We will also need to sample a coupling for four probability mass functions on $\{1, \ldots, N\}$ and this is presented in algorithm 8.

To continue onwards, we will consider a generalization of that in algorithm 4. The driving CCPF at level $l$ is described in algorithm 10. Now given algorithms 7–10, we are in a position to give our Markov kernel, $\check{K}^l : \Theta \times \mathsf{Z}^l \times \mathsf{Z}^{l-1} \to \mathcal{P}(\mathsf{Z}^l \times \mathsf{Z}^{l-1})$, which we shall call the coupled-CCPF (C-CCPF) and it is given in algorithm 11. To assist the subsequent discussion, we will introduce the marginal Markov kernel

$$
\check{q}_\theta^{(l)}\left([x_0^l, x_0^{l-1}], d[x_{\Delta_l:1}^l, x_{\Delta_{l-1}:1}^{l-1}]\right)
$$

$$
:= \int_{(\mathbb{R}^d)^{\Delta_l^{-1}} \times (\mathbb{R}^d)^{\Delta_{l-1}^{-1}}} \check{q}_\theta^l\left([(x_0^l, \bar{x}_0^l), (x_0^{l-1}, \bar{x}_0^{l-1})], d[(x_{\Delta_l:1}^l, \bar{x}_{\Delta_l:1}^l), (x_{\Delta_{l-1}:1}^{l-1}, \bar{x}_{\Delta_{l-1}:1}^{l-1})]\right). \tag{3.6}
$$

Given this kernel, one can describe the CCPF at two different levels $l, l-1$ in algorithm 9. Algorithm 9 details a Markov kernel $\check{C}^l : \Theta \times \mathsf{X}^l \times \mathsf{X}^{l-1} \to \mathcal{P}(\mathsf{X}^l \times \mathsf{X}^{l-1})$ which we will use in the initialization of our Markov chain to be described below.

We will consider the Markov chain $\{\check{Z}_m^l\}_{m \in \mathbb{N}_0}$, with

$$
\check{Z}_m^l = \left((X_{\Delta_l:n}^l(m), \bar{X}_{\Delta_l:n}^l(m)), (X_{\Delta_{l-1}:n}^{l-1}(m), \bar{X}_{\Delta_{l-1}:n}^{l-1}(m))\right),
$$

generated by the C-CCPF kernel in algorithm 11 and with initial distribution

$$
\check{\nu}_\theta^l\left(d[(x_{\Delta_l:n}^l, \bar{x}_{\Delta_l:n}^l), (x_{\Delta_{l-1}:n}^{l-1}, \bar{x}_{\Delta_{l-1}:n}^{l-1})]\right)
$$

$$
= \int_{\mathsf{Z}^l \times \mathsf{Z}^{l-1}} \left(\prod_{k=1}^n \check{q}_\theta^{(l)}\left([x_{k-1}^{l,\prime}, x_{k-1}^{l-1,\prime}], d[x_{k-1+\Delta_l:k}^{l,\prime}, x_{k-1+\Delta_{l-1}:k}^{l-1,\prime}]\right)\right)
$$

$$
\times \left(\prod_{k=1}^n \check{q}_\theta^{(l)}\left([\bar{x}_{k-1}^{l,\prime}, \bar{x}_{k-1}^{l-1,\prime}], d[\bar{x}_{k-1+\Delta_l:k}^{l,\prime}, \bar{x}_{k-1+\Delta_{l-1}:k}^{l-1,\prime}]\right)\check{C}_\theta^l\left([x_{\Delta_l:n}^{l,\prime}, x_{\Delta_{l-1}:n}^{l-1,\prime}], d[x_{\Delta_l:n}^l, x_{\Delta_{l-1}:n}^{l-1}]\right)\right)
$$

$$
\times \delta_{\{\bar{x}_{\Delta_l:n}^{l,\prime}, \bar{x}_{\Delta_{l-1}:n}^{l-1,\prime}\}}(d[\bar{x}_{\Delta_l:n}^l, \bar{x}_{\Delta_{l-1}:n}^{l-1}]), \tag{3.7}
$$

where $x_0^{l,\prime} = x_0^{l-1,\prime} = \bar{x}_0^{l,\prime} = \bar{x}_0^{l-1,\prime} = x_\star$. An important point, as in the case of algorithm 5, is that if the two input trajectories in step 1 of algorithm 11 are equal, i.e. $x_{\Delta_l:n}^l = \bar{x}_{\Delta_l:n}^l$, or $x_{\Delta_{l-1}:n}^{l-1} = \bar{x}_{\Delta_{l-1}:n}^{l-1}$, then the associated output trajectories will also be equal. As before, we define the stopping times associated with the given Markov chain $(\check{Z}_m^l)_{m \in \mathbb{N}_0}$, $s \in \{l, l-1\}$

$$
\tau^s = \inf\{m \geq 1 : X_{\Delta_s:n}^s(m) = \bar{X}_{\Delta_s:n}^s(m)\}.
$$

Then, setting $m^* \in \{2, 3, \ldots\}$ one has the following estimator:

$$
\widehat{\pi}_\theta^s(G_\theta^s) := G_\theta^s(X_{\Delta_s:n}^s(m^*)) + \sum_{m=m^*+1}^{\tau^s-1} \{G_\theta^s(X_{\Delta_s:n}^s(m)) - G_\theta^s(\bar{X}_{\Delta_s:n}^s(m))\}, \tag{3.8}
$$

and one sets $\Xi_\theta^l = \widehat{\pi}_\theta^l(G_\theta^l) - \widehat{\pi}_\theta^{l-1}(G_\theta^{l-1})$. The procedure for computing $\Xi_\theta^l$ is summarized in algorithm 12.

## (d) Estimate and remarks

Given the commentary above we are ready to present the procedure for our unbiased estimate of $\mathfrak{H}_\theta^{(ij)}$ for each $(i,j) \in D_\theta := \{(i,j) \in \{1, \ldots, d_\theta\}^2 : i \leq j\}$; $\mathfrak{H}_\theta$ is a symmetric matrix. The two main algorithms we will use (algorithms 6 and 12) are stated in terms of providing $\Xi_\theta^l$ for one specified function $G_\theta^{l,(i)}$ (recall that $i$ was suppressed from the notation). However, the algorithms can be run once and provide an unbiased estimate of $\pi_\theta^l(G_\theta^{l,(i)}) - \pi_\theta^{l-1}(G_\theta^{l-1,(i)})$ for every $i \in \{1, \ldots, d_\theta\}$, of

$\pi_\theta^l(G_\theta^{l,(i)} G_\theta^{l,(j)}) - \pi_\theta^{l-1}(G_\theta^{l-1,(i)} G_\theta^{l-1,(j)})$ and $\pi_\theta^l(H_\theta^{l,(ij)}) - \pi_\theta^{l-1}(H_\theta^{l-1,(ij)})$ for every $(i,j) \in D_\theta$. To that end, we will write $\Xi_\theta^l(G_\theta^{l,(i)})$, $\Xi_\theta^l(G_\theta^{l,(i)} G_\theta^{l,(j)})$ and $\Xi_\theta^l(H_\theta^{l,(ij)})$ to denote the appropriate estimators.

Our approach consists of the following steps, repeated for $k \in \{1, \ldots, M\}$:

(i) Generate $(L_k, \tilde{L}_k)$ according to $\mathbb{P}_L \otimes \mathbb{P}_L$.

(ii) Compute $\Xi_\theta^{k,0}(G_\theta^{0,(i)})$ for every $i \in \{1, \ldots, d_\theta\}$ and $\Xi_\theta^{k,0}(G_\theta^{0,(i)} G_\theta^{0,(j)})$, $\Xi_\theta^{k,0}(H_\theta^{0,(ij)})$ for every $(i,j) \in D_\theta$ using algorithm 6. Independently, compute $\tilde{\Xi}_\theta^{k,0}(G_\theta^{0,(i)})$ for every $i \in \{1, \ldots, d_\theta\}$ using algorithm 6.

(iii) If $L_k > 0$ then independently for each $l \in \{1, \ldots, L_k\}$ and independently of step 2, calculate $\Xi_\theta^{k,l}(G_\theta^{l,(i)})$ for every $i \in \{1, \ldots, d_\theta\}$ and $\Xi_\theta^{k,l}(G_\theta^{l,(i)} G_\theta^{l,(j)})$, $\Xi_\theta^{k,l}(H_\theta^{l,(ij)})$ for every $(i,j) \in D_\theta$ using algorithm 12.

(iv) If $\tilde{L}_k > 0$ then independently for each $l \in \{1, \ldots, \tilde{L}_k\}$ and independently of steps 2 and 3, calculate $\tilde{\Xi}_\theta^{k,l}(G_\theta^{l,(i)})$ for every $i \in \{1, \ldots, d_\theta\}$ using algorithm 12.

(v) Compute for every $(i,j) \in D_\theta$

$$\widehat{\mathfrak{H}}_\theta^{k,(ij)} = \Big( \sum_{l=0}^{L_k} \frac{\Xi_\theta^{k,l}(G_\theta^{l,(i)})}{\overline{\mathbb{P}}_L(l)} \Big) \Big( \sum_{l=0}^{\tilde{L}_k} \frac{\tilde{\Xi}_\theta^{k,l}(G_\theta^{l,(i)})}{\overline{\mathbb{P}}_L(l)} \Big) - \sum_{l=0}^{L_k} \frac{\Xi_\theta^{k,l}(G_\theta^{l,(i)} G_\theta^{l,(j)})}{\overline{\mathbb{P}}_L(l)}$$
$$- \sum_{l=0}^{L_k} \frac{\Xi_\theta^{k,l}(H_\theta^{l,(ij)})}{\overline{\mathbb{P}}_L(l)}.$$

Then our estimator is for each $(i,j) \in D_\theta$

$$\widehat{\mathfrak{H}}_\theta^{(ij)} = \frac{1}{M} \sum_{k=1}^{M} \widehat{\mathfrak{H}}_\theta^{k,(ij)}. \tag{3.9}$$

The algorithm and the various settings are described and investigated in detail in [22] as well as enhanced estimators. We do not discuss the methodology further in this work.

**Remark 3.3.** As we will see in the succeeding section, we will compare our methodology which is based on the C-CCPF to that of another methodology, which is the $\Delta$PF, within particle Markov chain Monte Carlo. Specifically, it will be a particle marginal Metropolis Hastings algorithm. We omit such a description of the latter, as we only use it as a comparison, but we refer the reader to [18] for a more concrete description. However, we emphasize that it is only asymptotically unbiased, in relation to the Hessian identity (2.4).

**Remark 3.4.** It is important to emphasize that with inverse Hessian, which is required for Newton methodologies, we can debias both the C-CCPF and the $\Delta$PF. This can be achieved by using the same techniques which are presented in the work of Jasra *et al.* [21].

## 4. Numerical experiments

In this section, we demonstrate that our estimate of the Hessian is unbiased through various different experiments. We consider testing this through the study of the variance and bias of the mean square error, while also providing plots related to the Newton-type learning. Our results will be demonstrated on three underlying diffusion processes: a univariate OU process, a multivariate OU process and the FHN model. We compare our methodology to that using the $\Delta$PF instead of the coupled-CCPF within our unbiased estimator.

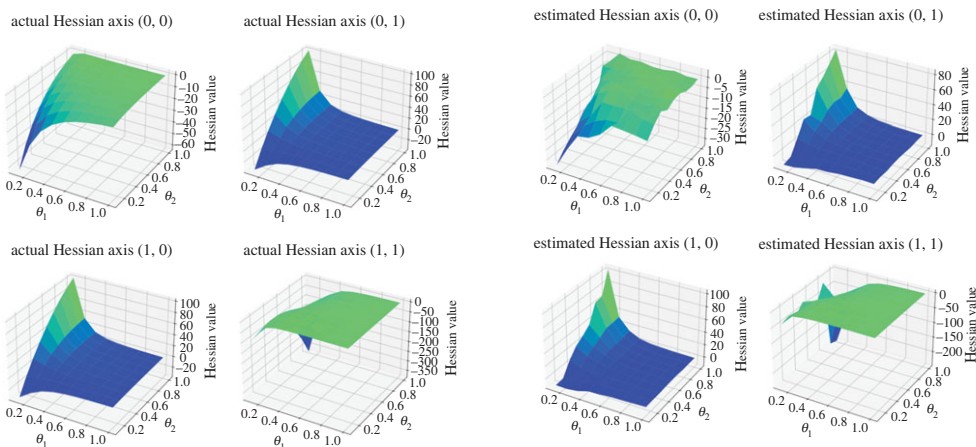

**Figure 2.** Experiments for the OU model. (*a*) True values of the Hessian. (*b*) Estimated values of the Hessian. (Online version in colour.)

## (a) Ornstein–Uhlenbeck process

Our first set of numerical experiments will be conducted on a univariate OU process, which takes the form

$$\mathrm{d}X_t = -\theta_1 X_t \, \mathrm{d}t + \sigma \, \mathrm{d}W_t$$

and

$$X_0 = x_0,$$

where $x_0 \in \mathbb{R}^+$ is our initial condition, $\theta_1 \in \mathbb{R}$ is a parameter of interest and $\sigma \in \mathbb{R}$ is the diffusion coefficient. For our discrete observations, we assume that we have Gaussian measurement errors, $Y_t | X_t \sim g_\theta(\cdot | X_t) = \mathcal{N}(X_t, \theta_2)$ for $t \in \{1, \dots, T\}$ and for some $\theta_2 \in \mathbb{R}^+$. Our observations will be generated with parameter choices defined as $\theta = (\theta_1, \theta_2) = (0.46, 0.38)$, $x_0 = 1$ and $T = 500$. Throughout the simulation, one observation sequence $\{Y_1, Y_2, \dots, Y_T\}$ is used. The true distribution of observations can be computed analytically, therefore the Hessian is known. In figure 2, we present the surface plots comparing the true Hessian with the estimated Hessian, obtained by the Rhee & Glynn estimator (3.9) truncated at discretization level $L = 8$, this is done by letting $\mathbb{P}_L(l) \propto \Delta_l \mathbb{I}(l \leq L)$. We use $M = 10^4$ to obtain the estimate Hessian surface plot. Both surface plots are evaluated at $\theta_1, \theta_2 \in \{0.2, 0.3, 0.4, \dots, 1.0\}$. In figure 3, we test out the convergence of bias of the Hessian estimate (3.9) with respect to its truncated discretization level. This essentially tests the result in lemma A.2. We use $L = \{2, 3, 4, 5, 6, 7\}$ and plot the bias against $\Delta_L$.

The bias is obtained by using $M = 10^4$ i.i.d. samples, and taking its entry-wise difference with the true Hessian entry-wise value. Note that the Hessian estimate here is evaluated with true parameter choice. As the parameter $\theta$ is two-dimensional, we present four log-log plots where the rate represents the fitted slope of log-scaled bias against log-scaled $\Delta_L$. We observe that the Hessian estimate bias is of order $\Delta_L^\alpha$ where $\alpha \in \{0.9629, 0.7536, 0.7361, 0.8949\}$ respectively for the four entries, which verifies our result in lemma A.2. We also compare the wall-clock time cost of obtaining one realization of Hessian estimate (3.9) with the cost of obtaining one realization of score estimate (see [22]), both truncated at the same discretization levels $L = \{2, 3, 4, 5, 6, 7\}$, here $M = 100$. The comparison result is provided in figure 4*a*.

We observe that the cost of obtaining the Hessian estimate is on average three times more expensive than obtaining a score estimate. The reason for this is that we need to simulate three CCPF paths in order to obtain one summand in the Hessian estimate, while to estimate the score function, we need only one path. We also record the fitted slope of log-scaled cost against log-scaled $\Delta_L$ for both estimates, the cost for Hessian estimates is roughly proportional

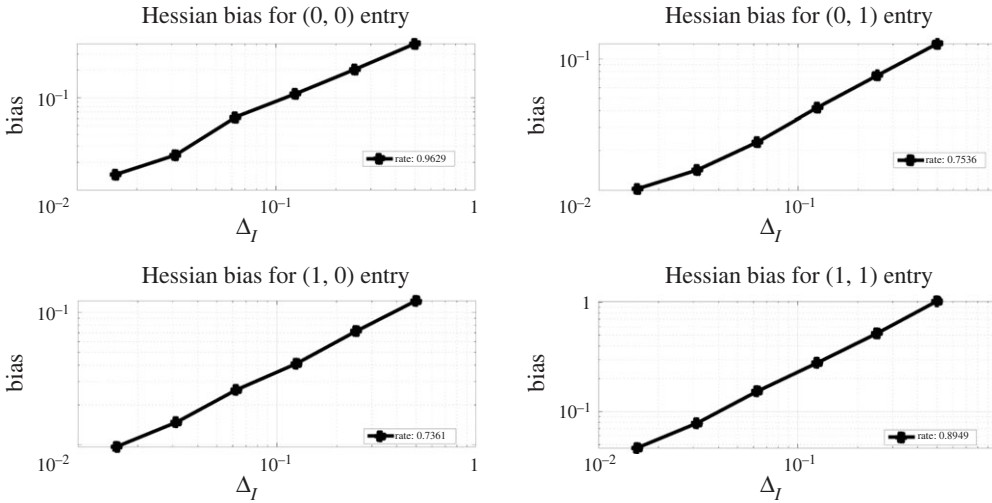

**Figure 3.** Experiments for the OU model: bias values of Hessian estimate. (Online version in colour.)

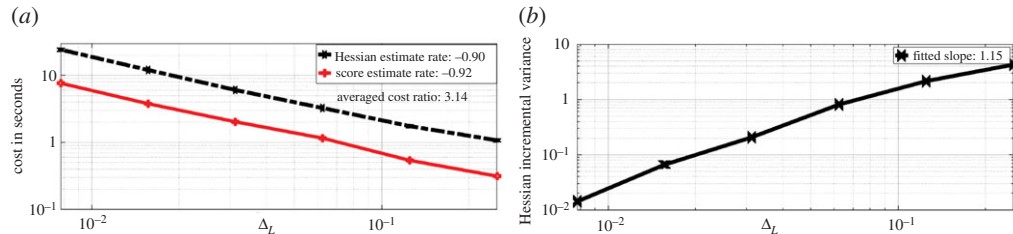

**Figure 4.** Experiments for the OU model. (*a*) Cost of Hessian & score estimate. (*b*) Incremental Hessian estimate variance summed over all entries. (Online version in colour.)

to $\Delta_L^{-0.9}$. To verify the rate obtained in lemma A.3, we compare the variance of the Hessian incremental estimate with respect to discretization level $L \in \{1, 2, 3, 4, 5, 6\}$. The incremental variance is approximated with the sample variance over $10^3$ repetitions, and we sum over all $2 \times 2$ entries and present the log-log plot of the summed variance against $\Delta_L$ on the right of figure 4. We observe that the incremental variance is of order $\Delta_L^{1.15}$ for the OU process model. This verifies the result obtained in lemma A.3. It is known that when truncated, the Rhee & Glynn method essentially serves as a variance reduction method. As a result, compared to the discrete Hessian estimate (2.6), the truncated Hessian estimate (3.9) will require less cost to achieve the same MSE target.

We present on figure 5*a*, the log-log plot of cost against MSE for discrete Hessian estimate (2.6) and the Rhee & Glynn (R&G) Hessian estimate (3.9). We observe that (2.6) requires much lower cost for an MSE target compared to (3.9). For (2.6), the cost is proportional to $\mathcal{O}(\epsilon^{-2.974})$ for an MSE target of order $\mathcal{O}(\epsilon^2)$. While for (3.9), the cost is proportional to $\mathcal{O}(\epsilon^{-2.428})$. The average cost ratio between (3.9) and (2.6) under the same MSE target is 5.605. In figure 5*b*, we present the log-log plot of cost against MSE for (3.9) and the Hessian estimate obtained by the $\Delta$PF method. We observe that under similar MSE target, the latter method on average has cost 5.054 times less than (3.9). In figure 6, we present the convergence plots for the stochastic gradient descent (SGD) method with score estimate and Newton method with score & Hessian estimate. For both methods, the parameter is initialized at $(0.1, 0.1)$, and the learning rate for the SGD method is set to 0.002. Our conclusion from figure 5*a* is that firstly the methodology using R&G has a better

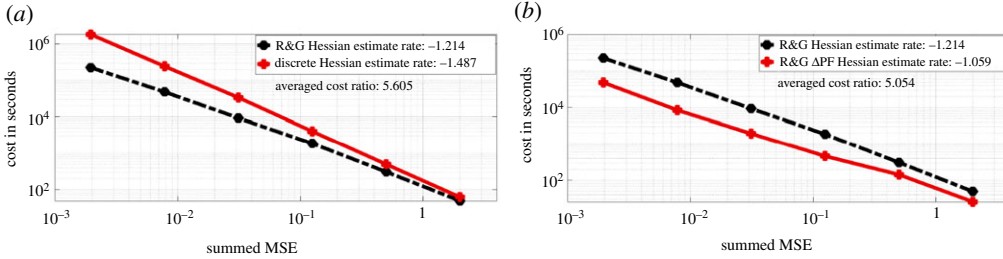

**Figure 5.** Hessian estimate cost against summed MSE for the OU model. (Online version in colour.)

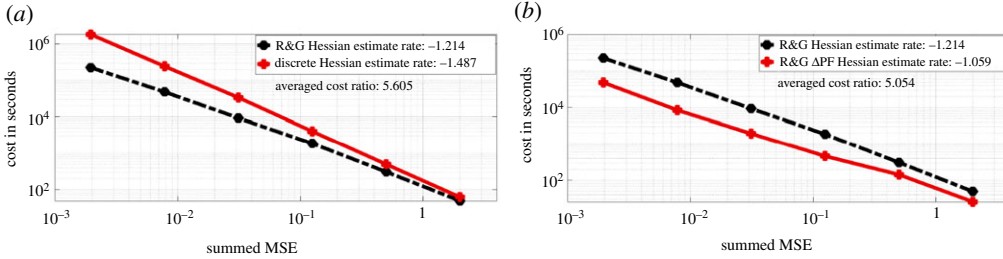

**Figure 6.** Parameter estimate for the OU model. (a) SGD with score estimate. (b) Newton method with score & Hessian estimate. (Online version in colour.)

rate, relating the MSE to computational cost, which favours our methodology. For figure 5b, we note that the methodology exploiting the $\Delta$ PF and our methodology, i.e. 'R&G Hessian', result in the same rate. As we know the former is unbiased, this comparison indicates our methodology is also unbiased, despite being more expensive by 5.054 times. For figure 6, we observe as expected that the Newton method requires fewer iterations (five compared to 132 which uses SGD) to reach the true parameters of interest, i.e. $\theta_1$ and $\theta_2$.

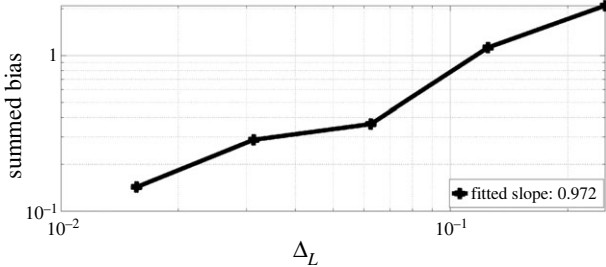

**Figure 7.** Hessian estimate bias summed over all entries for the multivariate OU diffusion model. (Online version in colour.)

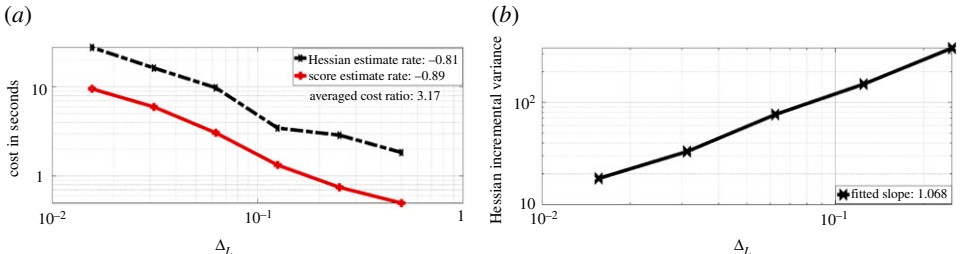

**Figure 8.** Experiments for the multivariate OU diffusion model. (*a*) Cost of Hessian and score estimate. (*b*) Incremental Hessian estimate variance summed over all entries. (Online version in colour.)

## (b) Multivariate Ornstein–Uhlenbeck process model

Our second model of interest is a two-dimensional OU process defined as

$$
\begin{bmatrix} dX_t^{(1)} \\ dX_t^{(2)} \end{bmatrix} = \begin{bmatrix} \theta_1 - \theta_2 X_t^{(1)} \\ -\theta_3 X_t^{(2)} \end{bmatrix} dt + \begin{bmatrix} \sigma_1 \\ \sigma_2 \end{bmatrix} dW_t, \quad X_0 = x_0,
$$

where $x_0 \in \mathbb{R}^2$ is the initial condition and $(\sigma_1, \sigma_2) \in \mathbb{R}^+ \times \mathbb{R}^+$ are the diffusion coefficients. We assume Gaussian measurement errors, $Y_t | X_t \sim g_\theta(\cdot | X_t) = \mathcal{N}_2(X_t, \theta_4 I_2)$, where $I_2$ is a two-dimensional identity matrix. We generate one sequence of observations up to time $T = 500$ with parameter choice $\theta = (\theta_1, \theta_2, \theta_3, \theta_4) = (0.48, 0.78, 0.37, 0.32)$, $\sigma_1 = 0.8$, $\sigma_2 = 0.6$, $x_0 = (1, 1)^T$. As before, we study various properties of (3.9) with the true parameter choice. In figure 7, we present the log-log plot of bias against $\Delta_L$ for (3.9), where the five points are evaluated with $L \in \{2, 3, 4, 5, 6\}$. The bias is approximated by the difference between (3.9) and the true Hessian with $M = 10^4$, we sum over all entry-wise bias and present it on the plot. We observe that the summed bias is of order $\Delta_L^{0.972}$. This verifies the result in lemma A.2. In figure 8*a*, we present a log-log plot of cost against $\Delta_L$ for (3.9) and the R&G score estimate both with $M = 10$. We observe that the cost of (3.9) is proportional to $\Delta_L^{-0.866}$. This rate is similar to that of the score estimate, on average the cost ratio between (3.9) and the score estimate is 3.495.

In figure 8, we present on the left the log-log plot of summed incremental variance of Hessian estimate against $\Delta_L$. We compute the entry-wise sample variance of the incremental Hessian estimate for $10^3$ times, and plot the summed variance against $\Delta_L$. We observe that the Hessian incremental variance is proportional to $\Delta_L^{1.068}$. This verifies the result in lemma A.3. In figure 9*a*, we present the log-log plot of the cost against MSE for (3.9) and (2.6), where the MSE is approximated through averaging over $10^3$ i.i.d. repetitions of both estimators. We observe that under a summed MSE target of $\mathcal{O}(\epsilon^2)$, the cost for (3.9) is of order $\mathcal{O}(\epsilon^{-2.362})$, while the cost for (2.6) is of order $\mathcal{O}(\epsilon^{-2.958})$. On average, the cost ratio between (2.6) and (3.9) is 3.575. This

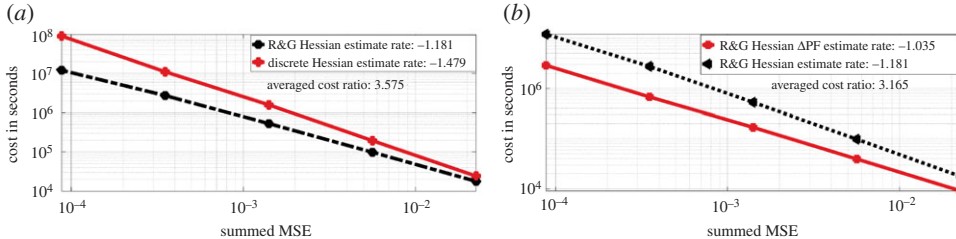

**Figure 9.** Hessian estimate cost against summed MSE for the multivariate OU diffusion model. (Online version in colour.)

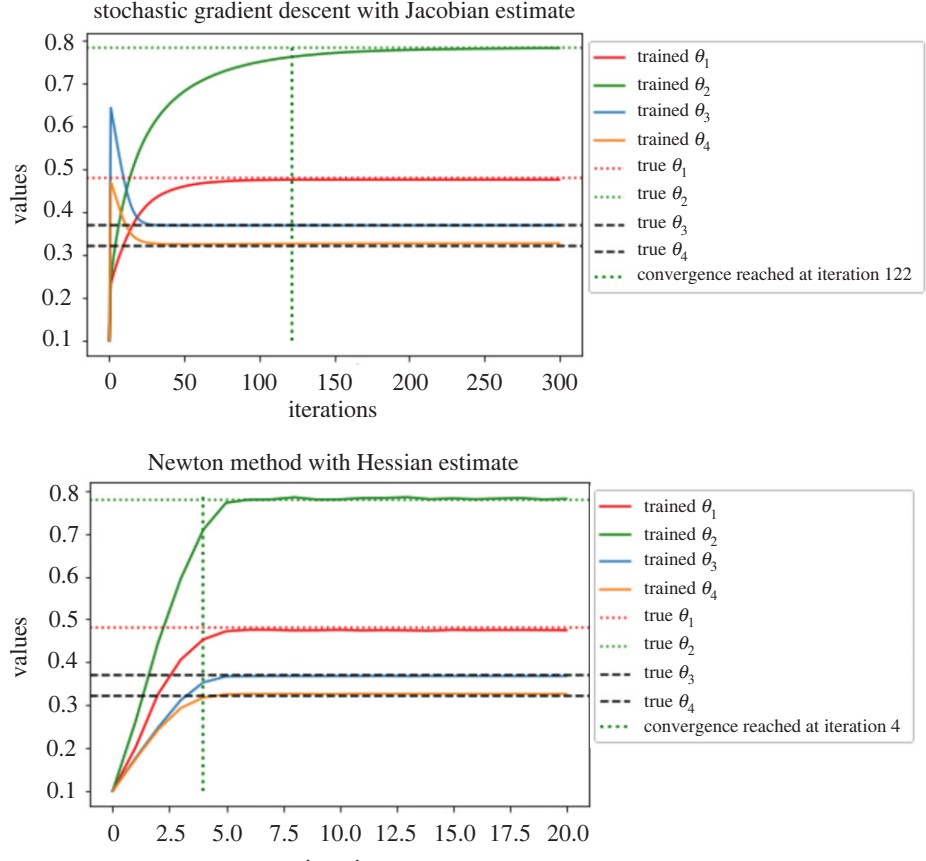

**Figure 10.** Parameter estimate for the multivariate OU model. (*a*) SGD with the score estimate. (*b*) Newton method with score and Hessian estimate. (Online version in colour.)

verifies the variance reduction effect of truncated R&G scheme. In figure 9*b*, we present the log-log plot of the cost against MSE for (3.9) and Hessian estimate using $\Delta$PF. We observe that under a similar MSE target, the latter method on average costs 3.165 times less than that of (3.9). In figure 10, we present the convergence plots for the SGD and Newton methods. Both the score estimate and the Hessian estimate (3.9) are obtained with $M = 2 \times 10^3$, truncated at level $L = 8$. The learning rate for the SGD is set to 0.005. The training reaches convergence when the relative Euclidean distance between trained and true $\theta$ is no bigger than 0.02. We initialize the training parameter at $(0.1, 0.1, 0.1, 0.1)$, and we observe that the SGD method reaches convergence with 122

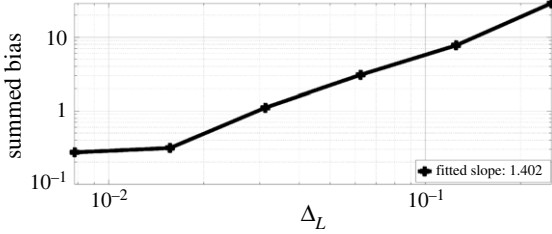

**Figure 11.** Hessian estimate bias summed over all entries for the FHN model.

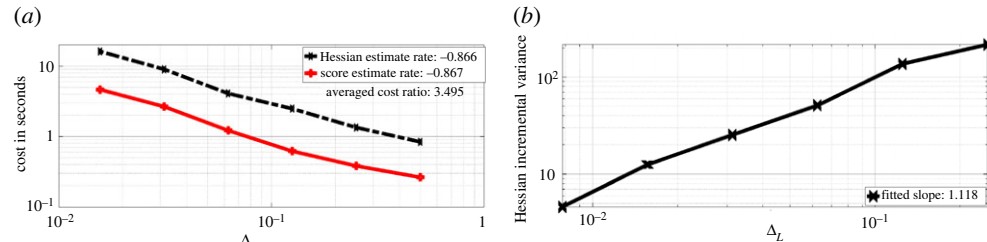

**Figure 12.** Experiments for the FHN model. (*a*) Cost of Hessian & score estimate. (*b*) Incremental Hessian estimate variance summed over all entries. (Online version in colour.)

iterations, compared to four iterations of the modified Newton method. The actual training time until convergence for the Newton method is roughly 7.6 times faster than the SGD method.

## (c) FitzHugh–Nagumo model

Our next model will be a two-dimensional ordinary differential equation, which arises in neuroscience, known as the FHN model [7,31]. It is concerned with the membrane potential of a neuron and a (latent) recovery variable modelling the ion channel kinetics. We consider a stochastic perturbed extended version, given as

$$
\begin{bmatrix} dX_t^{(1)} \\ dX_t^{(2)} \end{bmatrix} = \begin{bmatrix} \theta_1(X_t^{(1)} - (X_t^{(1)})^3 - X_t^{(2)}) \\ \theta_2 X_t^{(1)} - X_t^{(2)} + \theta_3 \end{bmatrix} dt + \begin{bmatrix} \sigma_1 \\ \sigma_2 \end{bmatrix} dW_t, \quad X_0 = u_0.
$$

For the discrete observations, we assume Gaussian measurement errors, $Y_t | X_t \sim g_\theta(\cdot | X_t) = \mathcal{N}_2(X_t, \theta_4 I_2)$, where $(\theta_1, \theta_2, \theta_3, \theta_4) \in \mathbb{R}^+ \times \mathbb{R} \times \mathbb{R} \times \mathbb{R}^+$, $(\sigma_1, \sigma_2) \in \mathbb{R}^+ \times \mathbb{R}^+$ are the diffusion coefficients and, as before, $\{W_t\}_{t \geq 0}$ is a Brownian motion. We generate one observation sequence with parameter choices $\theta = (\theta_1, \theta_2, \theta_3, \theta_4) = (0.89, 0.98, 0.5, 0.79)$, $\sigma = (0.2, 0.4)$. As the true distribution of the observation is not available analytically, we use $L = 10$ to simulate out $\{Y_1, Y_2, \ldots, Y_T\}$ where $T = 500$.

In figure 11, we compared the bias of (3.9), truncated at discretization level $L \in \{2, 3, 4, 5, 6, 7\}$ and plot it against $\Delta_L$ (log-log plot). The summed bias is obtained by taking element-wise difference between an average of $10^3$ i.i.d. realizations of the Hessian estimate and the true Hessian, then summed over all the element-wise difference. The true Hessian is approximated by (3.9) with $M = 10^4$ and $L = 10$. We observe that the summed bias is of order $\mathcal{O}(\Delta_L^{1.402})$. This verifies the result in lemma A.2. In figure 12*a*, we present the log-log plot of cost against $\Delta_L$ for (3.9) and the R&G score estimate both with $M = 10$. We observe that the cost of (3.9) is of order $\mathcal{O}(\Delta_L^{-0.866})$, while the cost for score estimate is of order $\mathcal{O}(\Delta_L^{-0.867})$. The average cost ratio between (3.9) and the score estimate is 3.495. In figure 12*b*, we present the log-log plot of the summed incremental

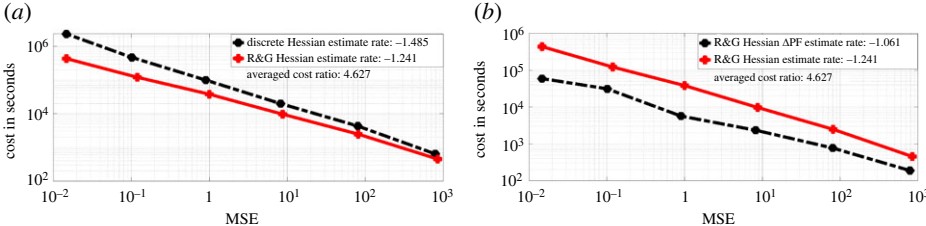

**Figure 13.** Hessian estimate cost against summed MSE for FHN model. (Online version in colour.)

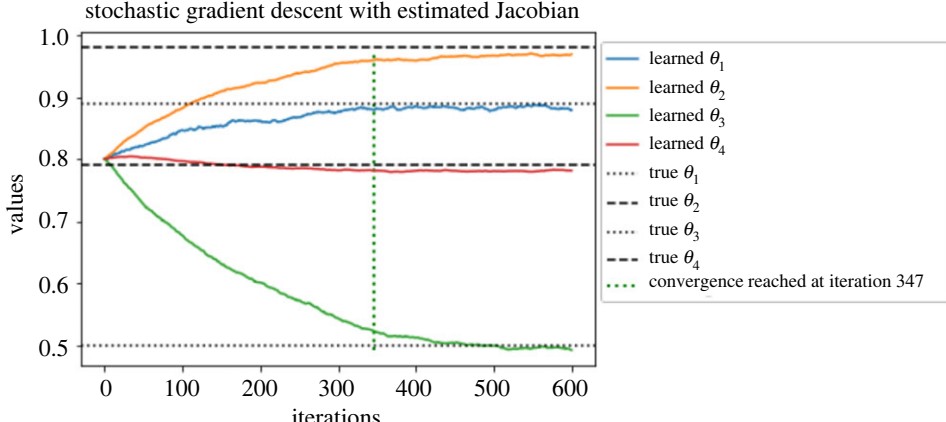

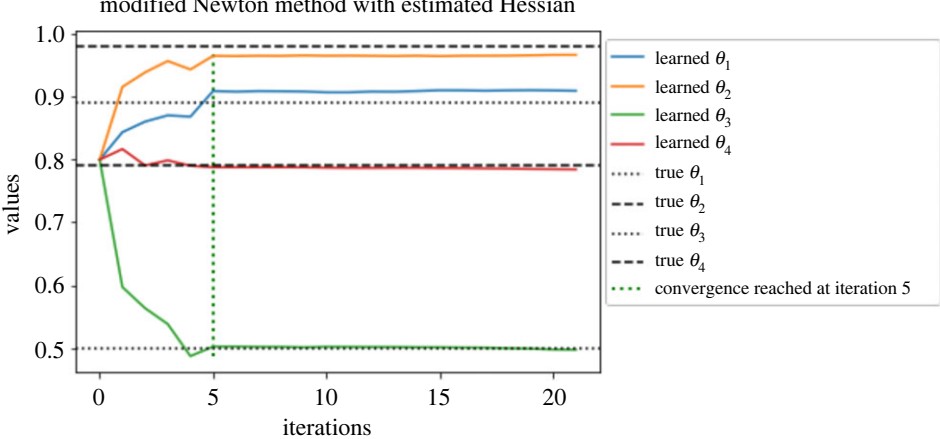

**Figure 14.** Parameter estimate for the FHN model. (*a*) SGD with score estimate. (*b*) Modified Newton method with score & Hessian estimate. (Online version in colour.)

variance with $\Delta_L$. We observe that the summed incremental variance is of order $\mathcal{O}(\Delta_L^{1.118})$. This verifies the result in lemma A.3.

In figure 13*a*, we again present a log-log plot of the cost against the summed MSE of (3.9) over all entries for both (3.9) and (2.6). We observe that under an MSE target of $\epsilon^2$, (3.9) requires cost of order $\mathcal{O}(\epsilon^{-2.482})$, while (2.6) requires cost of order $\mathcal{O}(\epsilon^{-2.97})$. The average cost ratio between (3.9) and (2.6) under the same MSE target is 3.987. This verifies the variance reduction effect of truncated R&G scheme. In figure 13*b* is the log-log plot of cost against summed MSE for (3.9) and Hessian estimate using the $\Delta$PF. We observe that under similar MSE target, the latter method on average costs 4.627 times less than that of (3.9). In figure 14, we present the convergence plots

of SGD and the modified Newton method. For the modified Newton method, we set all the off-diagonal entries to zero for the Hessian estimate, and add 0.0001 to the diagonal entries to avoid singularity. When the $L_2$ norm of the score is smaller than 0.1, we scale the searching step by a learning rate of 0.002. Both the score estimate and the Hessian estimate (3.9) are obtained with $M = 2 \times 10^3$, truncated at level $L = 8$. The learning rate for the SGD is set to 0.001. The training reaches convergence when the relative Euclidean distance between trained and true $\theta$ is no bigger than 0.02. We initialize the training parameter at $(0.8, 0.8, 0.8, 0.8)$, and observe that the SGD method reaches convergence with fewer iterations than the stochastic Newton method.

## 5. Summary

In this work, we were interested in developing an unbiased estimator of the Hessian, related to PODPs. This task is of interest, as computing the Hessian is primarily biased, due to its computational cost, but also it has improved convergence over the score function. We presented a general expression for the Hessian and proved, in the limit of discretization level, that it is consistent with the continuous form. We demonstrated that we were able to reduce the bias, arising from the discretization. This was shown through various numerical experiments that were tested on a range of diffusion processes. This not only highlighted the reduction in bias, but also that convergence is better compared to computing and using the score function. In terms of research directions beyond what we have done, it would be nice firstly to extend this to more complicated diffusion models, such as ones arising in mathematical finance [32,33]. Such diffusion models would be rough volatility models. Another potential direction would be to consider diffusion bridges, and analyse how one can could use the tools here and adapt them. This has been of interest, with recent works such as [34,35]. One could also aim to derive similar results for unbiased estimation using alternative discretization schemes, such as the Milstein scheme. This should result in different rates of convergence, however the analysis would be different. To do so, one would also require such a newly developed analysis for the score function [22]. Finally, one could aim to apply this to other applications, which are Monte Carlo methods being exploited such as phylogenetics. In particular, as we have tested our methodology on various OU processes, one could test this further on an SDE arising in phylogenetics, presented ad discussed in [36,37].

Data accessibility. Data and code for the paper can be found at https://github.com/fangyuan-ksgk/Hessian_Estimate.

Authors' contributions. N.K.C.: conceptualization, investigation, methodology, resources, supervision, writing—original draft, writing—review and editing; A.J.: formal analysis, methodology, project administration, resources, supervision, writing—original draft, writing—review and editing; F.Y.: resources, software, visualization.

All authors gave final approval for publication and agreed to be held accountable for the work performed therein.

Conflict of interest declaration. We declare we have no competing interest.

Funding. This work was supported by KAUST baseline funding.

## Appendix A. Proofs for proposition 2.1

In this section, we will consider a diffusion process $\{X_t^x\}_{t\geq 0} = \mathbf{X}_T^x$ which follows (2.1) and has an initial condition $X_0 = x \in \mathbb{R}^d$ and we will also consider Euler discretizations (2.5), at some given level $l$, which are driven by the same Brownian motion as $\{X_t^x\}_{t\geq 0}$ and the same initial condition, written $(\widetilde{X}_{\Delta_l}^x, \widetilde{X}_{2\Delta_l}^x, \ldots)$. We also consider another diffusion process $\{X_t^{x_\star}\}_{t\geq 0}$ which also follows (2.1), initial condition $X_0 = x_\star \in \mathbb{R}^d$ with the same Brownian motion as $\{X_t^x\}_{t\geq 0}$ and associated Euler discretizations, at level $l$, which are driven by the same Brownian motion as $\{X_t^{x_\star}\}_{t\geq 0}$ and the same initial condition, written $(\widetilde{X}_{\Delta_l}^{x_\star}, \widetilde{X}_{2\Delta_l}^{x_\star}, \ldots)$. The purpose of signifying the initial condition will be made apparent later on in the appendix. The expectation operator for the described process is written $\mathbb{E}_\theta$.

We require the following additional assumption called (D2) and all derivatives are assumed to be well defined.

— $[\Sigma^{-1}]^{j,k} \in \mathcal{B}_b(\mathbb{R}^d) \cap \mathrm{Lip}_{||\cdot||_2}(\mathbb{R}^d)$ $(j,k) \in \{1,\dots,d\}^2$.
— For any $\theta \in \Theta$, $a_\theta^j \in \mathcal{B}_b(\mathbb{R}^d)$, $\sigma^{j,k} \in \mathcal{B}_b(\mathbb{R}^d)$, $(j,k) \in \{1,\dots,d\}^2$.
— For any $\theta \in \Theta$, there exists $0 < \underline{C} < \overline{C} < +\infty$ such that for any $(x,y) \in \mathbb{R}^d \times \mathbb{R}^{d_y}$, $\underline{C} \leq g_\theta(y|x) \leq \overline{C}$. In addition for any $(\theta, y) \in \Theta \times \mathbb{R}^{d_y}$, $g_\theta(y|\cdot) \in \mathrm{Lip}_{||\cdot||_2}(\mathbb{R}^d)$.
— For any $(\theta, y) \in \Theta \times \mathbb{R}^{d_y}$, $\frac{\partial}{\partial \theta^{(i)}}(\log\{g_\theta(y|\cdot)\}) \in \mathcal{B}_b(\mathbb{R}^d) \cap \mathrm{Lip}_{||\cdot||_2}(\mathbb{R}^d)$, $i \in \{1,\dots,d_\theta\}$.
— For any $\theta \in \Theta$,

$$\frac{\partial}{\partial \theta^{(i)}}(b_\theta^{(j)}), \frac{\partial}{\partial \theta^{(i)}}(b_\theta^{(j)})^2, \frac{\partial^2}{\partial \theta^{(i)} \partial \theta^{(k)}}(b_\theta^{(j)}), \frac{\partial^2}{\partial \theta^{(i)} \partial \theta^{(k)}}(b_\theta^{(j)})^2 \in \mathcal{B}_b(\mathbb{R}^d) \cap \mathrm{Lip}_{||\cdot||_2}(\mathbb{R}^d)$$

$(i,k,j) \in \{1,\dots,d_\theta\}^2 \times \{1,\dots,d\}$.

The following result is proved in [22] and is lemma 1 of that article.

**Lemma A.1.** *Assume (D1-2). Then for any* $(n,r,\theta,i) \in \mathbb{N} \times [1,\infty) \times \Theta \times \{1,\dots,d_\theta\}$*, there exists a* $C < \infty$ *such that for any* $(l,x) \in \mathbb{N}_0 \times \mathbb{R}^d$

$$\mathbb{E}_\theta[|G_\theta^l(X_T^{l,x})^{(i)} - G_\theta(X_T^x)^{(i)}|^r]^{1/r} \leq C\Delta_l^{1/2}.$$

We have the following result which can be proved using very similar arguments to lemma A.1.

**Lemma A.2.** *Assume (D1-2). Then for any* $(n,r,\theta,i,j) \in \mathbb{N} \times [1,\infty) \times \Theta \times \{1,\dots,d_\theta\}^2$*, there exists a* $C < \infty$ *such that for any* $(l,x) \in \mathbb{N}_0 \times \mathbb{R}^d$:

$$\mathbb{E}_\theta[|\varphi_\theta(\widetilde{X}_{1:n}^x)G_\theta^l(X_T^{l,x})^{(i)} - \varphi_\theta(X_{1:n}^x)G_\theta(X_T^x)^{(i)}|^r]^{1/r} \leq C\Delta_l^{1/2},$$

$$\mathbb{E}_\theta[|\varphi_\theta(\widetilde{X}_{1:n}^x)H_\theta^l(X_T^{l,x})^{(ij)} - \varphi_\theta(X_{1:n}^x)H_\theta(X_T^x)^{(ij)}|^r]^{1/r} \leq C\Delta_l^{1/2}$$

$$\text{and} \quad \mathbb{E}_\theta[|\varphi_\theta(\widetilde{X}_{1:n}^x)G_\theta^l(X_T^{l,x})^{(i)}G_\theta^l(X_T^{l,x})^{(j)} - \varphi_\theta(X_{1:n}^x)G_\theta(X_T^x)^{(i)}G_\theta(X_T^x)^{(j)}|^r]^{1/r} \leq C\Delta_l^{1/2}.$$

*Proof of proposition 2.1.* In the following proof, we will suppress the initial condition from the notation. We have

$$\mathfrak{H}_\theta^{l,(ij)} - \mathfrak{H}_\theta^{(ij)} = \sum_{j=1}^3 T_j,$$

where

$$T_1 = \frac{\mathbb{E}_\theta[\varphi_\theta(\tilde{X}_{1:n})G_\theta^l(\mathbf{X}_T^l)^{(i)}]}{\mathbb{E}_\theta[\varphi_\theta(\tilde{X}_{1:n})]} \frac{\mathbb{E}_\theta[\varphi_\theta(\tilde{X}_{1:n})G_\theta^l(\mathbf{X}_T^l)^{(j)}]}{\mathbb{E}_\theta[\varphi_\theta(\tilde{X}_{1:n})]}$$

$$- \frac{\mathbb{E}_\theta[\varphi_\theta(X_{1:n})G_\theta(\mathbf{X}_T)^{(i)}]}{\mathbb{E}_\theta[\varphi_\theta(X_{1:n})]} \frac{\mathbb{E}_\theta[\varphi_\theta(X_{1:n})G_\theta(\mathbf{X}_T)^{(j)}]}{\mathbb{E}_\theta[\varphi_\theta(X_{1:n})]},$$

$$T_2 = \frac{\mathbb{E}_\theta[\varphi_\theta(\tilde{X}_{1:n})G_\theta^l(\mathbf{X}_T^l)^{(i)}G_\theta^l(\mathbf{X}_T^l)^{(j)}]}{\mathbb{E}_\theta[\varphi_\theta(\tilde{X}_{1:n})]} - \frac{\mathbb{E}_\theta[\varphi_\theta(X_{1:n})G_\theta(\mathbf{X}_T)^{(i)}G_\theta(\mathbf{X}_T)^{(j)}]}{\mathbb{E}_\theta[\varphi_\theta(X_{1:n})]}$$

$$\text{and} \quad T_3 = \frac{\mathbb{E}_\theta[\varphi_\theta(\tilde{X}_{1:n})H_\theta^l(\mathbf{X}_T^l)^{(ij)}]}{\mathbb{E}_\theta[\varphi_\theta(\tilde{X}_{1:n})]} - \frac{\mathbb{E}_\theta[\varphi_\theta(X_{1:n})H_\theta(\mathbf{X}_T)^{(ij)}]}{\mathbb{E}_\theta[\varphi_\theta(X_{1:n})]}.$$

We remark that

$$|\mathbb{E}_\theta[\varphi_\theta(\widetilde{X}_{1:n})] - \mathbb{E}_\theta[\varphi_\theta(X_{1:n})]| \leq C\Delta_l^{1/2}, \tag{A 1}$$

by using (D2) and convergence of Euler approximations of diffusions. For $T_1$, we have

$$T_1 = \left( \frac{\mathbb{E}_\theta[\varphi_\theta(\tilde{X}_{1:n}) G_\theta^l(\mathbf{X}_T^l)^{(i)}]}{\mathbb{E}_\theta[\varphi_\theta(\tilde{X}_{1:n})]} - \frac{\mathbb{E}_\theta[\varphi_\theta(X_{1:n}) G_\theta(\mathbf{X}_T)^{(i)}]}{\mathbb{E}_\theta[\varphi_\theta(X_{1:n})]} \right) \frac{\mathbb{E}_\theta[\varphi_\theta(\tilde{X}_{1:n}) G_\theta^l(\mathbf{X}_T^l)^{(j)}]}{\mathbb{E}_\theta[\varphi_\theta(\tilde{X}_{1:n})]}$$

$$+ \frac{\mathbb{E}_\theta[\varphi_\theta(X_{1:n}) G_\theta(\mathbf{X}_T)^{(i)}]}{\mathbb{E}_\theta[\varphi_\theta(X_{1:n})]} \left( \frac{\mathbb{E}_\theta[\varphi_\theta(\tilde{X}_{1:n}) G_\theta^l(\mathbf{X}_T^l)^{(j)}]}{\mathbb{E}_\theta[\varphi_\theta(\tilde{X}_{1:n})]} - \frac{\mathbb{E}_\theta[\varphi_\theta(X_{1:n}) G_\theta(\mathbf{X}_T)^{(j)}]}{\mathbb{E}_\theta[\varphi_\theta(X_{1:n})]} \right).$$

So one can easily deduce by ([22], theorem 1) and (D2) that for some $C$ that does not depend upon $l$

$$T_1 \le C\Delta_l^{1/2}.$$

Now for any real numbers, $a, b, c, d$ with $b, d$ non-zero, we have the simple identity

$$\frac{a}{b} - \frac{c}{d} = \frac{a}{bd}[d - b] + \frac{1}{d}[a - c].$$

So for $T_2, T_3$ combining this identity with lemma A.2, (A 1) and (D2) one can easily conclude that for some $C$ that does not depend upon $l$

$$\max\{T_2, T_3\} \le C\Delta_l^{1/2}.$$

From here, the proof is easily concluded. ∎

We end the section with a couple of results which are more-or-less direct corollaries of ([22], remarks 1 & 2). We do not prove them.

**Lemma A.3.** *Assume (D1–D2). Then for any $(i, j, r, \theta) \in \{1, \dots, d_\theta\}^2 \times [1, \infty) \times \Theta$, there exists a $C < \infty$ such that for any $(l, x, x_\star) \in \mathbb{N} \times \mathbb{R}^{2d}$*

$$\mathbb{E}_\theta[|H_\theta^l(\mathbf{X}_T^{l,x})^{(ij)} - H_\theta^{l-1}(\mathbf{X}_T^{l-1,x_\star})^{(ij)}|^r]^{1/r} \le C\left( \Delta_l^{1/2} + ||x - x_\star||_2 \right)$$

$$\mathbb{E}_\theta[|G_\theta^l(\mathbf{X}_T^{l,x})^{(i)} G_\theta^l(\mathbf{X}_T^{l,x})^{(j)} - G_\theta^{l-1}(\mathbf{X}_T^{l-1,x_\star})^{(i)} G_\theta^{l-1}(\mathbf{X}_T^{l-1,x_\star})^{(j)}|^r]^{1/r}$$

$$\le C\left( \Delta_l^{1/2} + ||x - x_\star||_2 \right).$$

# Appendix B. Algorithms

---

**Algorithm 4.** Driving Coupled Conditional Particle Filter at level 0.

---

1. Input $(x_{1:n}', \bar{x}_{1:n}') \in Z^0$. Set $k=1$, $(x_0^i, \bar{x}_0^i) = (x_\star, x_\star)$, $a_0^i = \bar{a}_0^i = i$ for $i \in \{1, \dots, N-1\}$.

2. Sampling: for $i \in \{1, \dots, N-1\}$ sample $(x_k^i, \bar{x}_k^i) | (x_{k-1}^{a_{k-1}^i}, \bar{x}_{k-1}^{\bar{a}_{k-1}^i})$ using the Markov kernel $\check{p}_\theta^0$ in Algorithm 2. Set $(x_k^N, \bar{x}_k^N) = (x_k', \bar{x}_k')$ and for $i \in \{1, \dots, N-1\}$, $(x_{1:k}^i, \bar{x}_{1:k}^i) = ((x_{1:k-1}^{a_{k-1}^i}, x_k^i), (\bar{x}_{1:k-1}^{\bar{a}_{k-1}^i}, \bar{x}_k^i))$. If $k=n$ stop.

3. Resampling: Construct the two probability mass functions on $\{1, \dots, N\}$:

$$r_1^i = \frac{g_\theta(y_k | x_k^i)}{\sum_{j=1}^N g_\theta(y_k | x_k^j)} \quad r_2^i = \frac{g_\theta(y_k | \bar{x}_k^i)}{\sum_{j=1}^N g_\theta(y_k | \bar{x}_k^j)} \quad i \in \{1, \dots, N\}.$$

For $i \in \{1, \dots, N-1\}$ sample $(a_k^i, \bar{a}_k^i)$ from the maximum coupling of the two given probability mass functions, using Algorithm 3. Set $k=k+1$ and return to the start of 2.

---

---

**Algorithm 5**. The Coupled Conditional Particle Filter at level 0.

---

1. Input $(x_{1:n}, \bar{x}_{1:n}) \in \mathsf{Z}^0$.
2. Run Algorithm 4.
3. Construct the two probability mass functions on $\{1, \dots, N\}$:

$$r_1^i = \frac{g_\theta(y_n | x_n^i)}{\sum_{j=1}^N g_\theta(y_n | x_n^j)} \quad r_2^i = \frac{g_\theta(y_n | \bar{x}_n^i)}{\sum_{j=1}^N g_\theta(y_n | \bar{x}_n^j)} \quad i \in \{1, \dots, N\}.$$

Sample $(i, j) \in \{1, \dots, N\}^2$ from the maximum coupling of the two given probability mass functions, using Algorithm 3. Return $(x_{1:n}^i, \bar{x}_{1:n}^j)$ which are the path of samples at indices $i$ and $j$ in step 2 of Algorithm 4 when $k = n$.

---

---

**Algorithm 6**. Computing $\varXi_\theta^0$.

---

1. Initialize the Markov chain by generating $Z_0$ using (3.4). Set $m = 1$
2. Generate $Z_m | Z_{m-1}$ using the Markov kernel described in Algorithm 5. If $x_{1:n}(m) = \bar{x}_{1:n}(m)$ stop and return $\varXi_0 = \widehat{\pi}_\theta^0(G_\theta^0)$ as in (3.5). Otherwise set $m = m + 1$ and return to the start of 2.

---

---

**Algorithm 7**. Simulating the Kernel $\check{q}_\theta^l$.

---

1. Input $(x_0^l, \bar{x}_0^l, x_0^{l-1}, \bar{x}_0^{l-1}) \in \mathbb{R}^{4d}$ and the level $l \in \mathbb{N}_0$.
2. Generate $V_{k\Delta_l} \overset{\text{i.i.d.}}{\sim} \mathcal{N}_d(0, \Delta_l I_d)$, for $k \in \{1, \dots, \Delta_l^{-1}\}$.
3. Run the two recursions, for $k \in \{1, \dots, \Delta_l^{-1}\}$:

$$X_{k\Delta_l}^l \&=\& X_{(k-1)\Delta_l}^l + a_\theta(X_{(k-1)\Delta_l}^l)\Delta_l + \sigma(X_{(k-1)\Delta_l}^l)V_{k\Delta_l}$$

$$\bar{X}_{k\Delta_l}^l \&=\& \bar{X}_{(k-1)\Delta_l}^l + a_\theta(\bar{X}_{(k-1)\Delta_l}^l)\Delta_l + \sigma(\bar{X}_{(k-1)\Delta_l}^l)V_{k\Delta_l}.$$

4. Run the two recursions, for $k \in \{1, \dots, \Delta_{l-1}^{-1}\}$:

$$X_{k\Delta_l}^{l-1} \&=\& X_{(k-1)\Delta_l}^{l-1} + a_\theta(X_{(k-1)\Delta_l}^{l-1})\Delta_l + \sigma(X_{(k-1)\Delta_l}^{l-1})[V_{(2k-1)\Delta_l} + V_{2k\Delta_l}]$$

$$\bar{X}_{k\Delta_l}^{l-1} \&=\& \bar{X}_{(k-1)\Delta_l}^{l-1} + a_\theta(\bar{X}_{(k-1)\Delta_l}^{l-1})\Delta_l + \sigma(\bar{X}_{(k-1)\Delta_l}^{l-1})[V_{(2k-1)\Delta_l} + V_{2k\Delta_l}].$$

5. Return $(x_{\Delta_l:1}^l, \bar{x}_{\Delta_l:1}^l, x_{\Delta_{l-1}:1}^{l-1}, \bar{x}_{\Delta_{l-1}:1}^{l-1}) \in \mathbb{R}^{\Delta_l^{-1}2d} \times \mathbb{R}^{\Delta_{l-1}^{-1}2d}$.

---

**Algorithm 8**. Simulating a Maximal Coupling of Maximal Couplings associated with Four Probability Mass Functions on $\{1, \ldots, N\}$.

1. Input: Four PMFs $(r_1^1, \ldots, r_1^N), \ldots, (r_4^1, \ldots, r_4^N)$ on $\{1, \ldots, N\}$.

2. If $r_1^i = r_3^i$ for every $i \in \{1, \ldots, N\}$ and there exists at least one $i \in \{1, \ldots, N\}$ such that $r_2^i \neq r_4^i$ then sample $(i_1, i_2)$ according to the maximal coupling of $(r_1^{i_1}, r_2^{i_2})$ in Algorithm 3. Implement 5. with $r_5^i = r_1^i$ and $r_6^i = r_4^i$, $i \in \{1, \ldots, N\}$ and $i_5 = i_1$. Set $(i_3, i_4) = (i_5, i_6)$ where $(i_5, i_6)$ have been computed from step 5. Go to 7.

3. If $r_2^i = r_4^i$ for every $i \in \{1, \ldots, N\}$ and there exists at least one $i \in \{1, \ldots, N\}$ such that $r_1^i \neq r_3^i$ then sample $(i_1, i_2)$ according to the maximal coupling of $(r_1^{i_1}, r_2^{i_2})$ in Algorithm 3. Implement 5. with $r_5^i = r_2^i$ and $r_6^i = r_3^i$, $i \in \{1, \ldots, N\}$ and $i_5 = i_2$. Set $(i_3, i_4) = (i_6, i_5)$ where $(i_5, i_6)$ have been computed from step 5. Go to 7.

4. Otherwise implement 6. with $r_{j+6}^i = r_j^i$, $(i, j) \in \{1, \ldots, N\} \times \{1, \ldots, 4\}$. Set $(i_1, \ldots, i_4) = (i_7, \ldots, i_{10})$ where $(i_7, \ldots, i_{10})$ have been computed from step 6. Go to 7.

5. Conditional Algorithm based on [38].

    (a) Input two PMFs $(r_5^1, \ldots, r_5^N), (r_6^1, \ldots, r_6^N)$ on $\{1, \ldots, N\}$ and $i_5 \in \{1, \ldots, N\}$ drawn according to $r_5$.

    (b) Sample $U \sim \mathcal{U}_{[0, r_5^{i_5}]}$. If $U < r_6^{i_5}$ set $i_6 = i_5$ and go to (c). Otherwise go to (b).

    (c) Sample $i_6'$ from $r_6^{i_6'}$. Sample $U' \sim \mathcal{U}_{[0, r_6^{i_6'}]}$. If $U' > r_5^{i_6'}$ set $i_6 = i_6'$ and go to (c). Otherwise start (b) again.

    (d) Output: $(i_5, i_6)$.

6. Sampling Maximal Couplings of Maximal Couplings

    (a) Input four PMFs $(r_7^1, \ldots, r_7^N), \ldots, (r_{10}^1, \ldots, r_{10}^N)$ on $\{1, \ldots, N\}$. For $j \in \{7, 9\}$ define the PMFs

$$\check{r}_j(i_j, i_{j+1}) = r_j^{i_j} \wedge r_{j+1}^{i_{j+1}} + \frac{r_j^{i_j} - r_j^{i_j} \wedge r_{j+1}^{i_j}}{1 - \sum_{i=1}^N r_j^i \wedge r_{j+1}^i} \left\{ r_{j+1}^{i_{j+1}} - r_j^{i_{j+1}} \wedge r_{j+1}^{i_{j+1}} \right\}.$$

    (b) Sample $(i_7, i_8)$ according to the maximal coupling of $(r_7^{i_7}, r_8^{i_8})$ in Algorithm 3. Generate $U \sim \mathcal{U}_{[0, \check{r}_7(i_7, i_8)]}$. If $U < \check{r}_9(i_7, i_8)$ set $(i_9, i_{10}) = (i_7, i_8)$ and go to (d). Otherwise go to (c).

    (c) Sample $(i_9', i_{10}')$ according to the maximal coupling of $(r_9^{i_9'}, r_{10}^{i_{10}'})$ in Algorithm 3. Sample $U' \sim \mathcal{U}_{[0, \check{r}_9(i_9', i_{10}')]}$. If $U' > \check{r}_7(i_9', i_{10}')$ set $(i_9', i_{10}') = (i_9, i_{10})$ and go to (d). Otherwise start (c) again.

    (d) Output: $(i_7, i_8, i_9, i_{10})$.

7. Output: $(i_1, i_2, i_3, i_4) \in \{1, \ldots, N\}^4$. $i_j$, marginally has PMF $r_j^i$, $j \in \{1, \ldots, 4\}$.

---

**Algorithm 9**. Coupled Conditional Particle Filter at level $l, l-1, l \in \mathbb{N}$.

1. Input $(x_{\Delta_l:n}', \bar{x}_{\Delta_{l-1}:n}') \in X^l \times X^{l-1}$. Set $k = 1$, $(x_0^i, \bar{x}_0^i) = (x_\star, x_\star)$, $a_0^i = \bar{a}_0^i = i$ for $i \in \{1, \ldots, N-1\}$.

2. Sampling: for $i \in \{1, \ldots, N-1\}$ sample $(x_{k-1+\Delta_l:k}^i, \bar{x}_{k-1+\Delta_{l-1}:k}^i) | (x_{k-1}^{a_{k-1}^i}, \bar{x}_{k-1}^{\bar{a}_{k-1}^i})$ using the Markov kernel $\check{q}_\theta^{(l)}$ in (3.6). Set $(x_{k-1+\Delta_l:k}^N, \bar{x}_{k-1+\Delta_{l-1}:k}^N) = (x_{k-1+\Delta_l:k}', \bar{x}_{k-1+\Delta_{l-1}:k}')$ and for $i \in \{1, \ldots, N-1\}$, $(x_{\Delta_l:k}^i, \bar{x}_{\Delta_{l-1}:k}^i) = ((x_{\Delta_l:k-1}^{a_{k-1}^i}, x_{k-1+\Delta_l:k}^i), (\bar{x}_{\Delta_{l-1}:k-1}^{\bar{a}_{k-1}^i}, \bar{x}_{k-1+\Delta_{l-1}:k}^i))$. If $k = n$ go to 4.

3. Resampling: Construct the two probability mass functions on $\{1, \ldots, N\}$:

$$r_1^i = \frac{g_\theta(y_k | x_k^i)}{\sum_{j=1}^N g_\theta(y_k | x_k^j)} \quad r_2^i = \frac{g_\theta(y_k | \bar{x}_k^i)}{\sum_{j=1}^N g_\theta(y_k | \bar{x}_k^j)} \quad i \in \{1, \ldots, N\}.$$

For $i \in \{1, \ldots, N-1\}$ sample $(a_k^i, \bar{a}_k^i)$ from the maximum coupling of the two given probability mass functions, using Algorithm 3. Set $k = k+1$ and return to the start of 2.

4. Construct the two probability mass functions on $\{1, \ldots, N\}$:

$$r_1^i = \frac{g_\theta(y_n | x_n^i)}{\sum_{j=1}^N g_\theta(y_n | x_n^j)} \quad r_2^i = \frac{g_\theta(y_n | \bar{x}_n^i)}{\sum_{j=1}^N g_\theta(y_n | \bar{x}_n^j)} \quad i \in \{1, \ldots, N\}.$$

Sample $(i, j) \in \{1, \ldots, N\}^2$ from the maximum coupling of the two given probability mass functions, using Algorithm 3. Return $(x_{\Delta_l:n}^i, \bar{x}_{\Delta_{l-1}:n}^j)$ which are the path of samples at indices $i$ and $j$ in step 2. when $k = n$.

---

**Algorithm 10**. Driving Coupled Conditional Particle Filter at level $l \in \mathbb{N}$.

---

1. Input $((x_{\Delta_l:n}^l, \bar{x}_{\Delta_l:n}^l), (x_{\Delta_{l-1}:n}^{l-1}, \bar{x}_{\Delta_{l-1}:n}^{l-1})) \in Z^l \times Z^{l-1}$. Set $k=1$, $(x_0^{i,l}, \bar{x}_0^{i,l})=(x_\star, x_\star)=(x_0^{i,l-1}, \bar{x}_0^{i,l-1})$, $a_0^{i,l}=\bar{a}_0^{i,l}=a_0^{i,l-1}=\bar{a}_0^{i,l-1}=i$ for $i \in \{1, \ldots, N-1\}$.

2. Sampling: for $i \in \{1, \ldots, N-1\}$ sample

$$\left((x_{k-1+\Delta_l:k}^{i,l}, \bar{x}_{k-1+\Delta_l:k}^{i,l}), (x_{k-1+\Delta_{l-1}:k}^{i,l-1}, \bar{x}_{k-1+\Delta_{l-1}:k}^{i,l-1})\right) \Big|$$

$$\left((x_{k-1}^{a_{k-1}^{i,l},l}, \bar{x}_{k-1}^{\bar{a}_{k-1}^{i,l},l}), (x_{k-1}^{a_{k-1}^{i,l-1},l-1}, \bar{x}_{k-1}^{\bar{a}_{k-1}^{i,l-1},l-1})\right)$$

using the Markov kernel $\check{q}_\theta^l$ in Algorithm 7. Set

$$\left((x_{k-1+\Delta_l:k}^{N,l}, \bar{x}_{k-1+\Delta_l:k}^{N,l}), (x_{k-1+\Delta_{l-1}:k}^{N,l-1}, \bar{x}_{k-1+\Delta_{l-1}:k}^{N,l-1})\right)=$$

$$\left((x_{k-1+\Delta_l:k}^l, \bar{x}_{k-1+\Delta_l:k}^l), (x_{k-1+\Delta_{l-1}:k}^{l-1}, \bar{x}_{k-1+\Delta_{l-1}:k}^{l-1})\right)$$

and for $i \in \{1, \ldots, N-1\}$

$$(x_{\Delta_l:k}^{i,l}, \bar{x}_{\Delta_l:k}^{i,l}) \&= \left((x_{\Delta_l:k-1}^{a_{k-1}^{i,l},l}, x_{k-1+\Delta_l:k}^{i,l}), (\bar{x}_{\Delta_l:k-1}^{\bar{a}_{k-1}^{i,l},l-1}, \bar{x}_{k-1+\Delta_l:k}^{i,l})\right),$$

$$(x_{\Delta_{l-1}:k}^{i,l-1}, \bar{x}_{\Delta_{l-1}:k}^{i,l-1}) \&= \left((x_{\Delta_{l-1}:k-1}^{a_{k-1}^{i,l-1},l}, x_{k-1+\Delta_{l-1}:k}^{i,l-1}), (\bar{x}_{\Delta_{l-1}:k-1}^{\bar{a}_{k-1}^{i,l-1},l-1}, \bar{x}_{k-1+\Delta_{l-1}:k}^{i,l-1})\right).$$

If $k=n$ stop.

3. Resampling: Construct the four probability mass functions on $\{1, \ldots, N\}$:

$$r_1^i = \frac{g_\theta(y_k|x_k^{i,l})}{\sum_{j=1}^N g_\theta(y_k|x_k^{j,l})} \quad r_3^i = \frac{g_\theta(y_k|\bar{x}_k^{i,l})}{\sum_{j=1}^N g_\theta(y_k|\bar{x}_k^{j,l})} \quad i \in \{1, \ldots, N\},$$

and

$$r_2^i = \frac{g_\theta(y_k|x_k^{i,l-1})}{\sum_{j=1}^N g_\theta(y_k|x_k^{j,l-1})} \quad r_4^i = \frac{g_\theta(y_k|\bar{x}_k^{i,l-1})}{\sum_{j=1}^N g_\theta(y_k|\bar{x}_k^{j,l-1})} \quad i \in \{1, \ldots, N\}.$$

For $i \in \{1, \ldots, N-1\}$ sample $(a_k^{i,l}, a_k^{i,l-1}, \bar{a}_k^{i,l}, \bar{a}_k^{i,l-1})$ using Algorithm 8. Set $k=k+1$ and return to the start of 2.

---

---

**Algorithm 11**. The Coupled-CCPF at level $l \in \mathbb{N}$.

---

1. Input $((x_{\Delta_l:n}^l, \bar{x}_{\Delta_l:n}^l), (x_{\Delta_{l-1}:n}^{l-1}, \bar{x}_{\Delta_{l-1}:n}^{l-1})) \in Z^l \times Z^{l-1}$.

2. Run Algorithm 10.

3. Construct the four probability mass functions on $\{1, \ldots, N\}$:

$$r_1^i = \frac{g_\theta(y_n|x_n^{i,l})}{\sum_{j=1}^N g_\theta(y_n|x_n^{j,l})} \quad r_3^i = \frac{g_\theta(y_n|\bar{x}_n^{i,l})}{\sum_{j=1}^N g_\theta(y_n|\bar{x}_n^{j,l})} \quad i \in \{1, \ldots, N\},$$

and

$$r_2^i = \frac{g_\theta(y_n|x_n^{i,l-1})}{\sum_{j=1}^N g_\theta(y_n|x_n^{j,l-1})} \quad r_4^i = \frac{g_\theta(y_n|\bar{x}_n^{i,l-1})}{\sum_{j=1}^N g_\theta(y_n|\bar{x}_n^{j,l-1})} \quad i \in \{1, \ldots, N\}.$$

Sample $(i_1, i_2, i_3, i_4) \in \{1, \ldots, N\}^4$ using Algorithm 8. Return $((x_{\Delta_l:n}^{i_1,l}, \bar{x}_{\Delta_l:n}^{i_3,l}), (x_{\Delta_{l-1}:n}^{i_2,l-1}, \bar{x}_{\Delta_{l-1}:n}^{i_4,l-1}))$ which are the path of samples at indices $i_{1:4}$ in step 2. of Algorithm 10 when $k=n$.

---

**Algorithm 12**. Computing $\varXi_\theta^l$.

1. Initialize the Markov chain by generating $\check{Z}_0^l$ using (3.7). Set $m=1$
2. Generate $\check{Z}_m^l|\check{Z}_{m-1}^l$ using the Markov kernel described in Algorithm 11. If $x_{\Delta_l:n}^l(m)=\bar{x}_{\Delta_l:n}^l(m)$ and $x_{\Delta_{l-1}:n}^{l-1}(m)=\bar{x}_{\Delta_{l-1}:n}^{l-1}(m)$ stop and return $\varXi_\theta^l=\hat{\pi}_\theta^l(G_\theta^l)-\hat{\pi}_\theta^{l-1}(G_\theta^{l-1})$ where $\hat{\pi}_\theta^s(G_\theta^s)$, $s\in\{l,l-1\}$, is as (3.8). Otherwise set $m=m+1$ and return to the start of 2.

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
