## [Peer Review File · Proceedings. Mathematical, Physical, and Engineering Sciences]

Review History

RSPA-2021-0710.R0 (Original submission)

Review form: Referee 1

Is the manuscript an original and important contribution to its field?

Good

Is the paper of sufficient general interest?

Good

Is the overall quality of the paper suitable?

Acceptable

Can the paper be shortened without overall detriment to the main message?

Yes

Do you think some of the material would be more appropriate as an electronic appendix?

No

Do you have any ethical concerns with this paper?

No

Recommendation?

Major revision is needed (please make suggestions in comments)

Comments to the Author(s)

The authors present an algorithmic estimation of the Hessian for partially observed diffusions. This is a necessary and not so obvious consideration of some of the authors' previous work. The manuscript is quite technical and written in a very dense way, where it may be very hard for an uninitiated reader to follow. I would recommend that the authors present an application first as a motivating argument, and through the application they set the stage for what it would follow. This would greatly help with the strategy part in Section 3a. Otherwise, the paper delves into a lot of detailed (yet necessary) remarks on stochastic calculus that could be perhaps avoided (or moved to the Appendix). Now, Section 3 lists a 12 Algorithms! that most of them are subroutines (necessary for the paper to be complete yet too distracting). I would strongly recommend that the authors present their major result which is the sampling of $\{X_i\}$'s (section 3c), and build out of there. Otherwise, the approach is very increasingly linear. A lot of subroutines could be added in the Appendix. Lastly, Section 4 is nice and offers a concrete application. The quality of all figures should be improved (tiny captions, headings etc), and as a comment, which could make the paper having a uniform story, is perhaps an example to phylogenetics, given that it is typical to employ an OU process therein e.g. see Jhwueng and Maroulas (2014). Lastly, the paper would greatly benefit from careful proofreading since it has several typos and grammatical errors.

Review form: Referee 2

Is the manuscript an original and important contribution to its field?

Good

Is the paper of sufficient general interest?

Good

Is the overall quality of the paper suitable?

Acceptable

Can the paper be shortened without overall detriment to the main message?

Yes

Do you think some of the material would be more appropriate as an electronic appendix?

No

Do you have any ethical concerns with this paper?

No

Recommendation?

Major revision is needed (please make suggestions in comments)

Comments to the Author(s)

See attached file (Appendix A).

Decision letter (RSPA-2021-0710.R0)

28-Jan-2022

Dear Dr Chada

The Editor of Proceedings A has now received comments from referees on the above paper and would like you to revise it in accordance with their suggestions which can be found below (not including confidential reports to the Editor).

Please submit a copy of your revised paper within four weeks - if we do not hear from you within this time then it will be assumed that the paper has been withdrawn. In exceptional circumstances, extensions may be possible if agreed with the Editorial Office in advance.

Please note that it is the editorial policy of Proceedings A to offer authors one round of revision in which to address changes requested by referees. If the revisions are not considered satisfactory by the Editor, then the paper will be rejected, and not considered further for publication by the journal. In the event that the author chooses not to address a referee's comments, and no scientific justification is included in their cover letter for this omission, it is at the discretion of the Editor whether to continue considering the manuscript.

To revise your manuscript, log into <http://mc.manuscriptcentral.com/prsa> and enter your Author Centre, where you will find your manuscript title listed under "Manuscripts with Decisions." Under "Actions," click on "Create a Revision." Your manuscript number has been appended to denote a revision.

You will be unable to make your revisions on the originally submitted version of the manuscript. Instead, revise your manuscript and upload a new version through your Author Centre.

When submitting your revised manuscript, you will be able to respond to the comments made by the referee(s) and upload a file "Response to Referees" in Step 1: "View and Respond to Decision Letter". Please provide a point-by-point response to the comments raised by the reviewers and the editor(s). A thorough response to these points will help us to assess your revision quickly. You can also upload a 'tracked changes' version either as part of the 'Response to reviews' or as a 'Main document'.

IMPORTANT: Your original files are available to you when you upload your revised manuscript. Please delete any unnecessary previous files before uploading your revised version.

When revising your paper please ensure that it remains under 28 pages long. In addition, any pages over 20 will be subject to a charge (£150 + VAT (where applicable) per page). Your paper has been ESTIMATED to be 24 pages.

Open Access

You are invited to opt for open access, our author pays publishing model. Payment of open access fees will enable your article to be made freely available via the Royal Society website as soon as it is ready for publication. For more information about open access please visit <https://royalsociety.org/journals/authors/open-access/>. The open access fee for this journal is £1700/\$2380/€2040 per article. VAT will be charged where applicable. Please note that if the corresponding author is at an institution that is part of a Read and Publishing deal you are required to select this option. See <https://royalsociety.org/journals/librarians/purchasing/read-and-publish/read-publish-agreements/> for further details.

Once again, thank you for submitting your manuscript to Proc. R. Soc. A and I look forward to receiving your revision. If you have any questions at all, please do not hesitate to get in touch.

Yours sincerely
 Raminder Shergill
 proceedingsa@royalsociety.org

on behalf of
 Professor Vincenzo Capasso
 Board Member
 Proceedings A

Reviewer(s)' Comments to Author:

Referee: 1

Comments to the Author(s)

The authors present an algorithmic estimation of the Hessian for partially observed diffusions. This is a necessary and not so obvious consideration of some of the authors' previous work. The manuscript is quite technical and written in a very dense way, where it may be very hard for an uninitiated reader to follow. I would recommend that the authors present an application first as a motivating argument, and through the application they set the stage for what it would follow. This would greatly help with the strategy part in Section 3a. Otherwise, the paper delves into a lot of detailed (yet necessary) remarks on stochastic calculus that could be perhaps avoided (or moved to the Appendix). Now, Section 3 lists a 12 Algorithms! that most of them are subroutines (necessary for the paper to be complete yet too distracting). I would strongly recommend that the authors present their major result which is the sampling of ξ 's (section 3c), and build out of there. Otherwise, the approach is very increasingly linear. A lot of subroutines could be added in the Appendix. Lastly, Section 4 is nice and offers a concrete application. The quality of all figures should be improved (tiny captions, headings etc), and as a comment, which could make the paper having a uniform story, is perhaps an example to phylogenetics, given that it is typical to employ an OU process therein e.g. see Jhwueng and Maroulas (2014). Lastly, the paper would greatly benefit from careful proofreading since it has several typos and grammatical errors.

Referee: 2

Comments to the Author(s)

see attached file

Board Member:

Comments to Author(s):

Based on the comments by the Reviewers, the authors are welcome to submit a thoroughly revised version of their manuscript.

The revised manuscript should be accompanied by a letter of detailed responses in the style Q-A (Question by reviewer - Answer by authors).

Author's Response to Decision Letter for (RSPA-2021-0710.R0)

See Appendix B.

RSPA-2021-0710.R1 (Revision)

Review form: Referee 1

Is the manuscript an original and important contribution to its field?

Good

Is the paper of sufficient general interest?

Good

Is the overall quality of the paper suitable?

Excellent

Can the paper be shortened without overall detriment to the main message?

Yes

Do you think some of the material would be more appropriate as an electronic appendix?

No

Do you have any ethical concerns with this paper?

No

Recommendation?

Accept as is

Comments to the Author(s)

The authors have successfully addressed all my comments, and in my opinion the paper is ready to be published.

Review form: Referee 2

Is the manuscript an original and important contribution to its field?

Good

Is the paper of sufficient general interest?

Good

Is the overall quality of the paper suitable?

Good

Can the paper be shortened without overall detriment to the main message?

Yes

Do you think some of the material would be more appropriate as an electronic appendix?

No

Do you have any ethical concerns with this paper?

No

Recommendation?

Accept as is

Comments to the Author(s)

The authors have answered by questions.

Decision letter (RSPA-2021-0710.R1)

23-Mar-2022

Dear Dr Chada

I am pleased to inform you that your manuscript entitled "Unbiased Estimation of the Hessian for Partially Observed Diffusions" has been accepted in its final form for publication in Proceedings A.

Our Production Office will be in contact with you in due course. You can expect to receive a proof of your article soon. Please contact the office to let us know if you are likely to be away from e-mail in the near future. If you do not notify us and comments are not received within 5 days of sending the proof, we may publish the paper as it stands.

As a reminder, you have provided the following 'Data accessibility statement' (if applicable). Please remember to make any data sets live prior to publication, and update any links as needed when you receive a proof to check. It is good practice to also add data sets to your reference list.
Statement (if applicable):

Open access

You are invited to opt for open access, our author pays publishing model. Payment of open access fees will enable your article to be made freely available via the Royal Society website as soon as it is ready for publication. For more information about open access please visit <https://royalsociety.org/journals/authors/which-journal/open-access/>. The open access fee for this journal is £1700/\$2380/€2040 per article. VAT will be charged where applicable.

Note that if you have opted for open access then payment will be required before the article is published – payment instructions will follow shortly.

If you wish to opt for open access then please inform the editorial office (proceedingsa@royalsociety.org) as soon as possible.

Your article has been estimated as being 27 pages long. Our Production Office will inform you of the exact length at the proof stage.

Proceedings A levies charges for articles which exceed 20 printed pages. (based upon approximately 540 words or 2 figures per page). Articles exceeding this limit will incur page charges of £150 per page or part page, plus VAT (where applicable).

Under the terms of our licence to publish you may post the author generated postprint (ie. your accepted version not the final typeset version) of your manuscript at any time and this can be made freely available. Postprints can be deposited on a personal or institutional website, or a recognised server/repository. Please note however, that the reporting of postprints is subject to a

media embargo, and that the status the manuscript should be made clear. Upon publication of the definitive version on the publisher's site, full details and a link should be added.

You can cite the article in advance of publication using its DOI. The DOI will take the form: 10.1098/rspa.XXXX.YYYY, where XXXX and YYYY are the last 8 digits of your manuscript number (eg. if your manuscript number is RSPA-2017-1234 the DOI would be 10.1098/rspa.2017.1234).

For tips on promoting your accepted paper see our blog post:
<https://royalsociety.org/blog/2020/07/promoting-your-latest-paper-and-tracking-your-results/>

On behalf of the Editor of Proceedings A, we look forward to your continued contributions to the Journal.

Sincerely,
Raminder Shergill
proceedingsa@royalsociety.org

on behalf of
Professor Vincenzo Capasso
Board Member
Proceedings A

Reviewer(s)' Comments to Author:

Referee: 1

Comments to the Author(s)

The authors have successfully addressed all my comments, and in my opinion the paper is ready to be published.

Referee: 2

Comments to the Author(s)

The authors have answered by questions.

Appendix A

Review on

Unbiased estimation of the hessian for partially observed diffusions

written by Chada, Jasra, Yu

The paper focuses on the estimation of the Hessian of the log-likelihood of a multi-dimensional diffusion process X_t of dimension d with observations Y which are noisy, discrete and partial observations of X_t . The log-likelihood of this model is generally not explicit, neither its Hessian. A common approach is to use discretization schemes, such as Euler-Maruyama scheme, which induce a discretization bias. The issue is thus to provide unbiased estimators. The authors propose to use a method introduced by Rhee and Glynn which randomizes on the level of the time-discretization within a MultiLevel Monte Carlo algorithm, with a coupling between the different levels. This approach has been used in a recent paper for score estimation of such models. The novelty of this paper is to apply it for the estimation of the Hessian.

My point of view on this paper is positive. I think there is an important need of developing new unbiased estimators for these models. The proposed method seems interesting, even if quite complex to implement.

Comments

- (1) Section Introduction. The problem is well presented but the novelty of the paper could be more emphasized. For example, the advantage of coupling strategies in this context could be more detailed. The advantage of computing the Hessian is not very clear as many estimation methods for PODP models are not based on gradient descent. Could this be justified?
- (2) p3 line 31: Let φ denotes \rightarrow Let φ denote
- (3) p4 line 30-31, please rephrase
- (4) p4 line 45: define $\mathbb{P}_\theta(\varphi_\theta(X_{t_1}, \dots, X_{t_n}))$
- (5) p5, line 41: why is this subsection numbered (i)?

- (6) p5, lines 43-50: Could you rephrase? The first density is called the joint density but it is a conditional density. Line 50, it is said to consider instead realizations of $(Y_{t_1}, \dots, Y_{t_n})$, but this is already the case in the previous lines. Clarify.
- (7) Section Diffusion Process. What are the limits of the hypothesis (i) and (ii)? The Fitzhug-Nagumo model used in the numerical section is not globally Lipschitz. Could the methodology be applied to locally Lipschitz model? Could hypoelliptic models be considered as well?
- (8) Section Time-discretization. The index ℓ should be defined. What is Section i mentioned line 48? It would more intuitive to write the transition density $p_\theta^\ell(\tilde{x}_p|\tilde{x}_{p-1})$ instead of the coma (as used later in Algorithm 1). Could you give the expression of this transition density where the intermediate points generated by the Euler scheme should play a role?
- (9) Proposition 2.1. The rate of convergence of the Euler scheme is only $\Delta^{1/2}$. Would it be interesting to use stronger order schemes to improve the speed of convergence of the algorithm?
- (10) Section Algorithm. The notations are heavy and not always consistent. Sometimes, the index l is used in particles $x_{\Delta_l:k}^{i,l}$, sometimes not. The Markov kernel p_θ^l is used to simulate intermediate points from x_{k-1} to the next observations time k , with time step Δ_l . This is not really said. Algorithm 1 mentions steps 2, 4 but they are denoted (ii), (iv).
- (11) P8 line 29, a variable Z_m is introduced. The link with x is only given p11. Why a special kernel is needed when $l = 0$? Could you give the intuition of the advantage of mixing several time steps of discretization? Could you control the rate of convergence of the final estimator depending on \mathbb{P}_L ?
- (12) A lot of algorithms are introduced. Maybe the presentations could be simplified by directly introducing a global algorithm?

- (13) Section Numerical experiments. Several typos (We uses, we needs, etc). The scales of the plots should be the same to ease the comparison (for e.g. Figure 2, Figure 5).
- (14) What is the conclusion of section (a)? What is the recommended algorithm? What is the gain in terms of parameter estimation of estimating the Hessian function? of estimating without bias this Hessian function?
- (15) Last sentence of Conclusion is not finished.

Appendix B

February 9, 2022

Prof. Vincenzo Capasso
Board Member, Proceedings of the Royal Society A:
Mathematical, Physical and Engineering Sciences

Dear Prof. Capasso,

Enclosed is a revised version of our manuscript, entitled “*Unbiased estimation of the hessian for partially observed diffusions*”. We appreciate the comments of the reviewers which have improved the manuscript, such as the clarity, presentation and the numerical experiments. We have made the appropriate changes according to the reviewers comments, highlighted in blue. Our responses are included directly after each comment.

Thanks again to the editor and reviewers for their input in reviewing this paper. We look forward to hearing about the status of our revision.

Sincerely,

Neil K. Chada
Ajay Jasra
Fangyuan Yu

Response to the first referee

We very much appreciate your careful reading, constructive suggestions and overall positive evaluation of our approach. We have followed all of your suggestions, and our point-to-point response to them follows.

Specific Major/General Comments

- 1) I would recommend that the authors present an application first as a motivating argument, and through the application they set the stage for what it would follow. This would greatly help with the strategy part in Section 3a. Otherwise, the paper delves into a lot of detailed (yet necessary) remarks on stochastic calculus that could be perhaps avoided (or moved to the Appendix).

We appreciate the reviewers comments related to the introduction. We firstly agree that an application would help motivate the work, before describing the details and procedures of the work. Given that we use a Fitzhugh-Nagumo model in the numerics, we decided to use this as a motivating example. This is complimented with a figure demonstrating how the dynamics behaves of this process. Furthermore we also believe that the main body of text, i.e. Section 3., is rather technical and long, primarily due to the number of algorithms. To make this easier for the reviewer have moved many of the algorithms to the appendix, which is related to the next point below.

- 2) Now, Section 3 lists a 12 Algorithms! that most of them are subroutines (necessary for the paper to be complete yet too distracting). I would strongly recommend that the authors present their major result which is the sampling of Ξ 's (section 3c), and build out of there. Otherwise, the approach is very increasingly linear. A lot of subroutines could be added in the Appendix.

Thank you for the comment. We agree with the reviewer that algorithm section is long and dense, which may not be easy to follow for a non-expert. As a result we have firstly decided to move most of the algorithms to the appendix, where we have cross-referenced them. We have also included our main result much earlier in Section 3, then before. From there we proceed with a discussion of the algorithms. It is important for us to explain each individually, for which we have kept this description largely the same. Again, the main difference is moving most of the algorithms to the appendix.

- 3) Lastly, Section 4 is nice and offers a concrete application. The quality of all figures should be improved (tiny captions, headings etc), and as a comment, which could make the paper having a uniform story, is perhaps an example to phylogenetics, given that it is typical to employ an OU process therein e.g. see Jhwueng and Maroulas (2014). Lastly, the paper would greatly benefit from careful proofreading since it has several typos and grammatical errors.

We thank the reviewer for his positive comments on Section 4. We have firstly improved this section by improving on the heading and axis labelling size, such that it is more readable, as well as some of the figure labels, and by making some of the figures bigger. We have also proofread the section, as we realized there were multiple typos and mistakes. These have now been modified correctly. Regarding the suggestion of using the phylogenetics example from Jhwueng and Maroulas (2014), we omit such an example as it is very much related to our initial examples of the OU process, where analytically one knows the solution. However we believe this area is of interest for practioners working with Monte Carlo methods, and remark on this potential application in the conclusion section. We have included the reference and another related one in the bibliography in the conclusion section.

Response to the second referee

We very much appreciate your careful reading, suggestions and overall positive evaluation of our approach. We have followed almost all of your suggestions, and our point-to-point response to them follows.

General Comments

- 1) **Section Introduction.** The problem is well presented but the novelty of the paper could be more emphasized. For example, the advantage of coupling strategies in this context could be more detailed. The advantage of computing the Hessian is not very clear as many estimation methods for PODP models are not based on gradient descent. Could this be justified?

It is not so obvious from the introduction why couplings helps in the context of our work. In general couplings can help produce unbiased estimators, which arises from the original work of [Rhee and Glynn 2014], where they use couplings to construct unbiased estimators of integrals with respect to an invariant distribution. Since this original work there has been numerous papers in this field, constructing different unbiased estimators, in different settings, motivated from the same work. As much of the work for unbiased estimation, related to filtering, has also exploited this methodology, we do the same here for our manuscript.

One advantage of this is how well this methodology can be applied to different problems, which is relatively simple to implement. The advantage of computing the Hessian, as we documented in the original submission, is that given the current computational power and modern advanced methodologies, one can exploit Hessian information for the improvement of the speed of convergence, compared to first order methodologies. The reviewer is correct that the Hessian has not been fully utilized in our setup, but recent work [Heng, et al. 2021] has used the same methodology for estimating the score function of PODPs, and other related papers. For us, considering the Hessian, for the reasons stated above, was a natural extension, especially for higher order gradient-based methods. We have made some of these reasons more clear, within the introduction section to help clarify on the novelty, and the use of couplings. We have also provided a few new references, which consider unbiased estimation through the coupling strategy, and in the context of the gradients of PODPs.

- 2)
 - p3 line 31: Let φ denotes \rightarrow Let φ denote.
 - p4 line 30-31, please rephrase.
 - p4 line 45: define $\mathbb{P}_\theta(\varphi_\theta(X_{t_1}, \dots, X_{t_n}))$.

This minor mistakes have been modified correctly.

- 3) p5, line 41: why is this subsection numbered (i)?

Thank you for finding this mistake, we put this initially as a subsubsection. To make it consistency we have now placed it as a subsection.

- 4) p5, lines 43-50: Could you rephrase? The first density is called the joint density but it is a conditional density. Line 50, it is said to consider instead realizations of $(Y_{t_1}, \dots, Y_{t_n})$, but this is already the case of the previous lines. Clarify.

We have rephrased both instances slightly. For the first point we have included the word “*conditional*”, but have kept the word “*joint*”. The reason for this is to emphasis that we are conditions

on the collection of y_{t_p} 's. For the latter point it is important that they are not the same as we specifically state we are using realizations of the random variables in the conditioning. We have made this more clear in Section 2(b).

- 5) **Section Diffusion Process.** What are the limits of the hypothesis (i) and (ii)? The Fitzhugh-Nagumo model used in the numerical section is not globally Lipschitz. Could the methodology be applied to locally Lipschitz model? Could hypoelliptic models be considered as well?

The assumptions we have placed for the diffusion processes are standard, which are uniform ellipticity, and a generic Lipschitz condition. The purpose of such assumptions is to ensure that solutions exist to such diffusion processes, there we can consider these standard and non-strong assumptions. The difficult with using hypoelliptic models is that, generally, Euler approximations don't work so well. Therefore one may require alternative discretization schemes. As we state later this is beyond the scope of such work, but is an interesting direction to consider. This would also result in an alternative analysis with different rates of convergence. This is also related to diffusions with non-globally Lipschitz conditions. Euler approximations do not work as well in this case, which as a result alternative schemes can be used such as a tamed-adaptive Euler-Maruyama approximation.

- 6) **Section Time-discretization.** The index ℓ should be defined. What is Section i mentioned line 48? It would more intuitive to write the transition density $p_\theta^\ell(\tilde{x}_p|\tilde{x}_{p-1})$ instead of the comma, (as used later in Algorithm 1). Could you give the expression of this transition density where the intermediate points generated by the Euler scheme should play a role?

We have clarified further what is l in the introduction of Section 2(e). We have removed the labelling of (i), therefore the referencing is now given as Section 2(d). We have also removed the comma notation and included $|$ instead. We have highlighted further what the transition density is, which is a density induced by the diffusion process (2.5). An exact expression can be given by writing a Normal density, with arbitrary mean and variance, which we have not specified, but further stated. The form the density would take is

$$p_\theta^\ell(\tilde{x}_p|\tilde{x}_{p-1}) = \frac{1}{\sqrt{2\pi\mathbb{V}[\tilde{x}_p]}} \exp\left(-\frac{(\tilde{x}_p - \mathbb{E}[\tilde{x}_p])^2}{2\mathbb{V}[\tilde{x}_p]}\right),$$

where $\mathbb{V}[\cdot]$ denotes the variance. Given how we have not specified the mean or the variance, we omit such an inclusion.

- 7) **Proposition 2.1.** The rate of convergence of the Euler scheme is only $\Delta^{1/2}$. Would it be interesting to use stronger order schemes to improve the speed of convergence of the algorithm?

We thank the reviewer for the comment. Indeed one could consider using alternative discretization schemes, that could result in alternative rates. Our initial reasons for choosing the Euler scheme is that firstly due to its simplicity, allowing analysis to be more easily derived, but previous results in unbiased estimation for diffusion processes have assumed Euler discretization for Δ . Despite this, we agree it would be of interest, and therefore we state this in the conclusion section, that this could potentially be future work to consider.

- 8) **Section Algorithm.** The notations are heavy and not always consistent. Sometimes, the index l used in the particles $x_{\Delta_l:k}^{i,l}$, sometimes not. The Markov kernel p_θ^l is used to simulate intermediate points from x_{k-1} to the next observations at time k , with time step Δ_l . This is not really said. Algorithm 1 mentions steps 2,4 but they are denoted as (ii) and (iv).

The notation of superscript l is used when a discretized x is used, or to denote couplings at

different levels. We have kept this consistent, however these ideas are not important when stating Algorithm 1, which is a generic conditional particle filter. It would be helpful to explicitly state the purpose of the Markov kernel. Therefore we have stated this before Remark 3.1. We agree the numbering of the steps, and the referencing is not correct within the algorithms. We have modified this, removing any use of roman numerals for each algorithms step.

- 9) P8 line 29, a variable Z_m is introduced. The link with x is only given p11. Why a special kernel is needed when $l = 0$? Could you give the intuition of the advantage of mixing several time steps of discretization? Could you control the rate of convergence of the final estimator depending on \mathbb{P}_L ?

The connection between Z_m and x is first provided on p8, so the same page. We have highlighted this in blue to make it clear where it is. Specifically, we define it as $\{Z_m\}_{m \in \mathbb{N}_0}$, where $Z_m = (X_{1:n}(m), \bar{X}_{1:n}(m))$.

In the case of $l = 0$: we deal with a single level only (see the development below). One can think of say a finite sum of differences of expectations and that one starts with a single expectation, of which one does not need to couple, but one needs to approximate. The level $l = 0$ could very well be any initial level (eg 3). The rate of convergence presumably is referred to in-terms of e.g. Monte Carlo error of the final estimator? In which case, if the estimator has a finite variance this is $\mathcal{O}(M^{-1})$ with M is the number of samples. This previously mentioned finite variance is only achievable if one chooses \mathbb{P}_L appropriately and as indicated in Proposition 3.1., but the rate of convergence is the Monte Carlo rate. Different \mathbb{P}_L can change the constant but not the rate.

The statement ‘mixing several time steps of discretization’ is unclear to us. The principle is essentially this. You have a sequence of probabilities $(\pi_l)_{l \in \mathbb{N}_0}$ with, for appropriate φ

$$\lim_{l \rightarrow \infty} \pi_l(\varphi) = \pi(\varphi),$$

for some fixed probability π . Then we note that

$$\lim_{l \rightarrow \infty} \pi_l(\varphi) = \sum_{l=1}^{\infty} \{\pi_l(\varphi) - \pi_{l-1}(\varphi)\} + \pi_0(\varphi).$$

The idea here is that one can approximate the infinite sum eg by sampling an integer L and using

$$\frac{1}{\mathbb{P}_L(l)} \widehat{\pi_l(\varphi) - \pi_{l-1}(\varphi)},$$

with $\widehat{\pi_l(\varphi) - \pi_{l-1}(\varphi)}$ an unbiased approximation of $\pi_l(\varphi) - \pi_{l-1}(\varphi)$. So you need the multiple discretization steps in the framework we adopt, to achieve unbiasedness.

- 10) A lot of algorithms are introduced. Maybe the presentations could be simplified by directly introducing a global algorithm?

We agree that the sheer quantity and content (in particular the notation) may seem overkill at that times, for the reader. To help make the paper more readable, we have moved most of the algorithms to the appendix where we have referred to them in the original sections. We have only kept some of the algorithms in Section 3. Producing a global algorithm would be cumbersome, therefore we believe a separation of the algorithms is better, as it explains each necessary step required.

- 11) **Section Numerical experiments. Several typos (We uses, we needs, etc). The scales of the plots should be the same to ease the comparison (for e.g. Figure 2, Figure 5).**

Thank you for comment. We have made the plots more presentable by making the axis labels and the legends more visible. Furthermore we have removed the typos, related to “we uses/needs” and corrected other found typos. Related to the scale plots, such as in Figure 2 and 5, we do not require the axis to be the same for Figure 2, as what is important is the rates, which are all relatively close to 1. For the parameter estimation plots, we consider an upper scale on the axis of the true parameter θ . To help indicate the performance, we originally included a “convergence after iteration” to clarify how quickly the parameter estimation occurs.

- 12) **What is the conclusion of section (a)? What is the recommended algorithm? What is the gain in terms of parameter estimation of estimating the Hessian function? of estimating without bias this Hessian function?**

We thank the reviewer for the comment. We forget the provide some conclusions on the last few figures for the OU process. From our findings in Figure 4 and 5, we can observe from the left subplot of Figure 4 that R&G methodology is preferred to the discrete case, as it attains a lower MSE-to-Cost rate. This helps justify, why these methods are also preferable in large-scale problems, where one requires a particular order of the MSE. For the right subplot of Figure 4, this indicates that our methodology attains, roughly, the same rates with the methodology implemented in Chada et al. [11], despite the later being more cost-effective and biased. As motivated throughout the paper, the Hessian can benefit the computational procedure by improving the convergence to the true unknown parameter. This is shown and verified in Figure 5, where we observe this. Figure 5 is not necessarily related to the fundamental idea of the paper, but acts as a motivation of why one would be interested in exploiting Hessian information for parameter estimation. Related to the bias comment, it is well-known that numerous parameter estimation problems use stochastic gradient methods, which are biased. Therefore ways to overcome this, while not sacrificing significant computational cost, can improve accuracy. To make these points more clear, we have clarified these points at the end of Section 4 (a).

- 13) **Last sentence of Conclusion is not finished.**

Thank you for spotting this, this was a typo which we meant to state earlier. We have removed this now.